# Cryoablation and Intratumoral Immunotherapy for Breast Cancer: A Future Path to Cost-Effective De-Escalation for Larger Tumors, Lymph Nodes and Metastatic Disease

**DOI:** 10.3390/cancers17121915

**Published:** 2025-06-09

**Authors:** Josephine Fermanian, Robert C. Ward, Dennis R. Holmes, Ariel C. Fisher, Jennifer Harvey, Brian Marples, Peter J. Littrup

**Affiliations:** 1School of Medicine and Health Sciences, George Washington University, Washington, DC 20052, USA; 2Department of Diagnostic Imaging, Rhode Island Hospital, Warren Alpert Medical School of Brown University, Providence, RI 02903, USA; rward@brownhealth.org; 3Department of Surgery, Adventist Health Glendale, Glendale, CA 91206, USA; dennis.holmes@xpeditemd.com; 4Department of Imaging Sciences, University of Rochester Medical Center, New York, NY 14642, USAjennifer_harvey@urmc.rochester.edu (J.H.); peter_littrup@urmc.rochester.edu (P.J.L.); 5Department of Radiation Oncology, University of Rochester Medical Center, New York, NY 14642, USA; brian_marples@urmc.rochester.edu

**Keywords:** breast cancer, cryoablation, de-escalation, immunotherapy, intratumoral

## Abstract

This review explores cryoablation as a promising, cost-effective option to de-escalate surgical treatment in breast cancer. Regrettably, its use is currently limited to small tumors under 1.5 cm in select patients, as larger tumors pose a greater risk of metastatic spread. Adjunctive therapies such as hormone therapy, radiation, and chemotherapy are still required for regional and systemic control but come with significant side effects and morbidities. Intravenous immunotherapy also has high associated morbidities. Direct tumor injections (intratumoral) of immunotherapy drugs may help reduce these risks. This review summarizes current evidence suggesting that strategically placed cryoablation probes combined with intratumoral immunotherapy may enhance treatment effectiveness, potentially offering improved protection against metastatic disease while reducing the complications and costs associated with traditional cancer therapies.

## 1. Introduction

Breast cancer remains a deadly problem; despite advances in chemotherapy, surgical resection, radiation, hormone therapy and targeted therapy, over 43,000 women in 2024 were estimated to die [1]. Cryoablation has been suggested as an option for the de-escalation of local surgical control in patients with low-biological-risk breast cancers (i.e., <1.5 cm) in order to reduce overall morbidity and cost, as well as improve quality of life, or the risk–benefit ratio [2,3,4,5,6,7,8,9,10,11]. This review covers the potential to treat larger breast tumors, lymph nodes and metastatic disease by combining cryoablation with intratumoral immunotherapy, while acknowledging current unknowns that require immediate, or near-term trials to better define.

De-escalation is a concept which embraces the spirit of the Hippocratic oath of “do no harm," by suggesting that the lower morbidity option be considered between cancer treatments when tumor response outcomes otherwise appear to be equivalent [12]. The exuberance for recent landmark immunotherapy outcomes may be warranted but still requires thorough evaluation of what are still unknowns. This review suggests that de-escalation using breast cryoablation should not be limited to the early-disease stage in elderly patients, or those with significant comorbidities and is proving to be more than just a limited tool for small cancers.

When we combine optimal and strategic local cryoablation techniques for the release of tumor antigens with optimized local immunotherapy (e.g., intratumoral injections of checkpoint inhibitors, etc.), the exceedingly low apparent side effects from adverse events may produce a rapid paradigm shift in our traditional perspectives of stage-based tumor treatments and may potentially de-escalate treatment for a larger percentage of breast cancer patients. This is already occurring with other tumors, such as stage II seminoma, which makes it more plausible [13]. Moreover, this review will conclude with an overview of cost estimates and patient quality of life considerations that may further highlight the necessity for cost-effective de-escalation strategies in breast cancer treatments.

### 1.1. A Spectrum of De-Escalation and Acceptance of Surgical Lumpectomy

De-escalation has perhaps received the most attention for its use with small, less aggressive breast cancers [2,3,4,5,6,7,8,9,10,11], but is expanding to other cancer types, such as testicular, thyroid, bladder and colon cancer [13]. For prostate cancer, conservative nontreatment of early tumors by “watchful waiting” has also stimulated the question of whether some breast cancers may similarly be observed for lower risk lesions found on core-needle biopsy [9]. With rapid advances in immunotherapy, marked changes in the frontline treatments have occurred within the last 12 months, such as those for urothelial cancers [14]. Similar rapid changes with the addition of immunotherapeutic drugs have not produced major changes in breast cancer treatments.

De-escalation has also occurred throughout the long history of surgical management of most breast cancers, extending back to the Halsted radical mastectomy in 1894 [15]. This debilitating extensive surgery was continually improved upon and eventually led to the landmark trials in 2002 showing that conservative approaches to breast cancer treatment are safe and equally, if not more, effective in relation to improved quality of life [16,17]. This also ushered in the routine use of radiation therapy in conjunction with lumpectomy, as well as chemotherapy and recent targeted agents [18]. This also extends to disease management of the axillary lymph nodes, particularly in combination with radiation therapy and/or neoadjuvant chemotherapy [19,20]. Immunotherapy for breast cancer has been predominantly limited to advanced stages in combination with various chemotherapies, using intravenous cytokines, checkpoint inhibitors and expensive cellular-based therapies, none of which have significantly changed or de-escalated upfront treatment protocols [21].

### 1.2. Breast Cancer Tumor Size and Metastatic Potential

Breast cancer occurs on a disease spectrum, from a local tumor, to regional (e.g., axillary nodes) and distant metastasis, whereby initial tumor size and cancer type affects the risk of disease spread. Appropriate tumor destruction using image-guided cryoablation as a local management tool can be optimized to improve tumor margin control for non-surgical patients with breast cancer tumors > 1.5 cm. Notably, axillary lymph node involvement is frequently present at the time of initial breast cancer diagnosis. Data from studies indicate that nodal metastases are found in roughly one-quarter of patients with tumors measuring 2 cm or less, increasing to over half of those with tumors that range between 2 and 5 cm, and nearly four out of five patients when tumors exceeding 5 cm. These trends suggest that lymphatic spread often occurs early in disease progression [22].

Although molecular subtypes affect prognosis, this plays less of a role in the overall prevalence of lymph node metastases, with 31% for ER+/HER2-, 34% for triple negative(ER-/PR-/HER2-) and 40% for HER2+ cancers. Similarly, more distant metastases were noted in up to 1.4% of patients with breast cancers < 2 cm, up to 10% for tumors 2–5 cm and up to ~25% for cancers > 5 cm. These greater biological risks of local breast recurrence, spread to nodes and/or distant metastatic sites for tumors > 1.5 cm have been historically addressed by adding additional therapies [16,17,18,19,20,21]. Table 1 illustrates the relationship of tumor size to stage of disease at diagnosis and clinical treatments which lead to opportunities for de-escalation due to their current associated morbidities.

From Table 1, we observe an accelerating risk of nodal involvement and metastasis with increasing tumor size. Systemic therapies, i.e., chemotherapy, targeted therapies (e.g., anti-HER2), and/or immunotherapy, are often administered for tumors > 1 cm, or >T1c. Recent studies suggest that replacing the toxicities of chemotherapy and radiation with combinations of cryoablation and localized intratumoral immunotherapy shows great potential. This approach may elicit a sufficient abscopal effect, not only targeting the local tumor but also clearing involved lymph nodes, and potentially offering sustained immune protection against further recurrence, whether it is nodal or systemic. While these strategies require further validation, their significant potential to reduce treatment-related morbidity and costs makes them highly promising for near-future clinical trials.

## 2. Current Adjunctive Therapies to Reduce Recurrence and Metastatic Disease

Despite aggressive surgical intervention, adjuvant therapies are recommended to reduce cancer recurrence and improve patient outcomes post-surgery [16,17,18,19,20]. Established commonly used adjunctive treatments including chemotherapy, hormonal therapy and radiation therapy (RT) will not be covered in-depth for this review. Notwithstanding advances in commonly used adjuvant treatments, microscopic cancer cells left behind in the margins and metastatic disease spread continue to be a wicked global clinical problem in the management of breast cancer [23]. This review briefly covers current generally accepted adjuvant therapies and complication rates to highlight the necessity of optimizing margin control with intratumoral ablation immunotherapy to potentially reduce metastatic disease while de-escalating complications and costs.

All adjuvant therapies are based on the type of breast cancer diagnosed; for example, hormonal therapy (HT) is recommended for patients with tumors that are hormone receptor-positive for a duration of five to ten years post-surgery [24,25]. Burstein and Griggs’s study found that adjuvant endocrine therapy reduces the risk of tumor recurrence by 40%; while it can potentially be life extending, patients frequently report reduced quality of life due to side effects and many prematurely discontinue HT.

While chemotherapy may have more toxicity than hormonal therapy, it also still addresses the high percentage of micro-metastatic disease as tumors become larger, especially for triple-negative and HER2-positive disease [26,27,28,29,30]. However, chemotherapy increases overall non-breast cancer mortality and the risk of heart failure is three times higher in people treated for breast cancer [30]. While taxanes, another class of chemotherapy drugs, work to prevent cancer cells dividing or mitosis, they also increase the risk of leukemia [30].

Radiation therapy (RT) is also a commonly used adjuvant treatment; however, tumor resistance in breast cancer is common [31,32]. Hufnagle et al.’s investigation into radiation-induced cardiac toxicity risk factors found major adverse cardiac events resulting from therapeutic radiation [33]. Moreover, exposure to ionizing radiation during radiotherapy for breast cancer increases the risk of ischemic heart disease [34,35,36,37].

## 3. Cryoablation

Investigations into the use of cryoablation, a method of destroying tissue by freezing, began in the mid-19th century when Dr. James Arnott discovered that crushed ice at −18 °C to −24 °C applied to breasts, cervical and skin cancers resulted in the shrinkage of tumors and a decrease in pain [38]. Cryoablation has a unique advantage over current adjuvant therapies because of its ability to produce thorough necrosis by disrupting the cell membrane while activating an immune response against distant tumors [39]. Cryoablation appears to work best in specific subtypes of breast cancer, particularly those that are early-stage, hormone receptor-positive (ER+/PR+), and HER2-negative (HER2-), especially tumors that are small, low-grade, and unifocal. These tumors offer the most consistent and favorable outcomes due to their less aggressive behavior and well-defined margins [4,7,10].

Cryoablation of breast tumors is normally performed in an outpatient setting, and most patients have reported the procedure as virtually painless due to the analgesic effect of the tumor freezing process. Breast cryoablation can be better understood from the perspective of benign and malignant [40,41,42,43,44,45,46] uses, along with the resultant types of cryoablation equipment that originally entered the market.

Cryoablation for benign breast fibroadenomas (FAs) received FDA approval in 2001, and was assigned a billing code in 2007, prompting two companies to develop compact cryoablation systems equipped with only a single cryoprobe tailored for outpatient clinics [42]. This was appropriate for benign conditions where FAs often have a central blood vessel, and its thorough destruction could be achieved with a single cryoprobe and ice extending just beyond the FA margins. This is different from generating lethal freezing temperatures beyond all cancer margins, which are usually between −20 to −40 °C for nearly all types of cancer cells (Figure 1) [43,47,48]. A single cryoprobe may be adequate for treating carefully selected small breast cancers. Multi-probe cryoablation, on the other hand, offers the potential for substantial treatment de-escalation in breast cancer, much like the transition from radical mastectomy to lumpectomy. This approach aims to provide comparable local and regional disease control to surgery, but with less associated morbidity [2,3,4,5,6,7,8,9,10,11]. This may be particularly applicable for elderly and non-surgical patients, as well as women who decline a surgical option.

Cryoablation has also been evaluated as a cost-effective treatment with improved quality-of-life advantages that are also not necessarily limited to <1.5 cm in these difficult-to-treat patient groups [43,44,45,46,47]. The feasibility of providing thorough cytotoxic isotherms extending beyond all local margins of breast cancers > 1.5 cm is thus readily obtainable with an appropriate cryoablation technique [43,47], but requires further considerations of adjunctive techniques for better control of local, nodal and distant metastatic recurrences (e.g., immunotherapy/intratumoral immunotherapy). Cryoablation’s efficacy relies on achieving cytotoxic isotherms that fully encompass the tumor, but as Littrup et al. (2009) emphasize, tissue heterogeneity, such as variations between dense and fatty breast tissue, can lead to uneven heat distribution and inconsistent ablation margins, potentially increasing the risk of local recurrence. Similarly, Huang et al. (2023) underscore the importance of precise cryoprobe placement and a tailored technique to compensate for the thermal conductivity differences in heterogeneous breast tissue to ensure complete tumor coverage and minimize recurrence risk [5,43]. Pusceddu et al. highlight the feasibility of retreating recurrent breast cancer with cryoablation. In their study, 35 patients with stage IV breast cancer underwent CT-guided cryoablation of the primary tumor. Over an average follow-up of 46 months, seven patients (20%) experienced local recurrences. Notably, all recurrences were successfully managed with additional cryoablation sessions, demonstrating that repeat cryoablation is a practical and effective option for managing local tumor recurrence [44]. Addressing the cancer margins requires a thorough understanding that multiple cryoprobes are needed for most tumors > 1.5 cm due to the simple heat load effect of body temperature tissues, making the quoted ice diameters from single cryoprobes much smaller than the relative lethal zone of two or more probes [49,50]. Such size-related sculpting of lethal ice to cover all apparent tumor margins produced no apparent local recurrences at the 18-month follow-up from one of the earliest cryoablation series without the excision of the breast cancers, which had an average tumor size of a 1.7 cm range of 0.5–5.8 cm [43]. Littrup et al. provide compelling evidence that cryoablation alone can offer effective local control for tumors larger than 1.5 cm. In their study of soft tissue tumors, including those exceeding 4 cm in diameter, over 80% of cases achieved successful local ablation with minimal complications and without the need for adjunct therapies. These results suggest that, with meticulous planning and appropriate imaging guidance, cryoablation can be a stand-alone treatment capable of sustained tumor control, even for larger lesions [48]. In some cases, CT-guidance has been crucial for the deeper cryoablation of tumors in lung, kidney and liver cancers [51,52,53], where many were only visible via CT.

### 3.1. The Use of Cytotoxic Isotherms and the Potential for Treating Larger Breast Tumors

The isotherm differences between current cryotechnologies for breast cancer appear to be minimal, using a single larger cryoprobe with a liquid nitrogen-based system (e.g., 3.4 mm diameter; IceCure Medical, Caesarea, 3079504, Israel) [49] versus the smaller, potentially multiple cryoprobes with a Joule–Thompson (JT) argon-based system (e.g., 2.1 mm diameter; Boston Scientific, Marlborough, MA, USA; Galil) [50]. The sizes of the 2 crucial isotherms at −20 °C and −40 °C are given over 10 min with widths and lengths of 33 × 36 mm and 24 × 28 mm, respectively, for the 3.4 mm probe with a liquid nitrogen-based system [49], compared with 32 × 49 mm and 23 × 42 mm with the 2.1 mm argon-based cryoprobes [50]. The longer freeze length of the JT argon gas 2.1 mm cryoprobe relates in part to its design for combining multiple cryoprobes to produce more spherical ice, whereas the liquid nitrogen probe has a shorter exposed length to similarly create spherical ice with a single larger probe. As we have previously noted, larger diameter probes have a larger surface area for better conductivity of the isotherms into the adjacent tissue [47], but the larger liquid nitrogen cryoprobes also achieve a faster initial icing surrounding the probe. Despite these potential benefits, the overall critical isotherms further out into the tissue are minimal. More importantly, strategic placement within the tumor and adjusting for the local heat sink are more critical than each cryoprobe’s isotherms (Figure 2) [43,47].

Figure 2 emphasizes the accuracy required for a single 2.4 mm cryoprobe to cover a hypothetical ~1.5 cm breast tumor, as well as how those isotherms are distorted by the greater heat sink and temperatures closer to the chest wall [43]. This also corroborates the previously noted greater positive breast cancer recurrence along the posterior ablation margin as reported in one of the first human breast cryoablation studies [40]. The anterior retraction technique noted below prevents not only the progression of the ice ball into the underlying pectoralis muscle when possible, but also reduces the impact of the chest wall heat sink. Monitoring posterior ice progression is not possible by ultrasound alone due to the intense shadowing of the anterior leading ice edge, but that posterior ice margin is more readily noted using intermittent CT fluoroscopy and helical acquisitions during phases of the freeze cycles (i.e., between the first and second freeze cycles and at the conclusion of the second freeze cycle).

Sabel et al. assessed patients with invasive ductal carcinoma (IDC) who underwent US-guided cryoablation for tumors ≤ 1.0 cm, showing the 100% destruction of cancer with a single thick cryoprobe (i.e., 2.4 mm outer diameter) [41]. Complete tumor destruction was obtained only for tumors between 1.0 and 1.5 cm without a significant ductal carcinoma in situ (DCIS) component. They concluded that cryoablation should be limited to a 1.5 cm diameter in IDC tumors [41]. However, the limit of <1.5 cm for the cryoablation of breast cancer appears to be relatively arbitrary from both old and new post-lumpectomy histology, when using a single cryoprobe to produce cytotoxic ice (e.g., <−30 °C) [40,41,54]. A more recent series [55] also used Joule–Thompson-based argon gas systems and similar size cryoprobe diameters (i.e., 2.1 mm) to produce 97% tumor eradication of ER+/HER2- tumors < 2 cm. Imaging follow-up in the post-cryoablation setting is critical to ensure satisfactory ablation targeting and coverage [55]. If suspicious imaging findings arise, biopsy should be performed to differentiate post-ablation change, such as fat necrosis, from tumor recurrence [54,55].

Figure 3 below is taken from the thermocouple-validated phantom experiments using CT monitoring [47] and it illustrates the impact of a probe on the resultant isotherms within the low-density ice. It also emphasizes the greater transference of the probe cooling to the adjacent tissues, proportionate to their probe diameters and associated surface areas over time (i.e., 5, 10 and 15 min).

Recent reports from the Ice3 trial [56,57] involving 194 subjects, with a mean age of 74.9 years and hormone receptor-positive, HER2/neu-negative invasive breast cancer < 1.5 cm (range: 2.5–14.9 mm; mean diameter 8.1 mm), showed an only 2.1% ipsilateral breast tumor recurrence rate at 3 years [56] and 4.3% at 5 years [57]. The study used a single 3.4 mm diameter cryoprobe, which provides an approximately 42% greater heat transfer capacity compared to a 2.4 mm diameter probe due to the difference in the surface area ratio [47]. In other words, excellent local outcomes of cryoablation for breast cancer can be obtained with appropriate attention to tumor size in relation to the underlying expected cytotoxic isotherms. We have demonstrated in several organ sites and different tumor types that thorough cytotoxic isotherms are not only related to tumor size but are more dependent on the placement and number of cryoprobes [47,48,49,50,51,52,53] and there is no significant advantage by their associated cryogens (i.e., argon gas versus liquid nitrogen).

### 3.2. Cryoablation Techniques: Imaging Guidance and the “Knuckle Rule”

As previously noted, CT-guidance augments the circumferential visualization of the ice margin in relation to adjacent crucial structures, whether that be underlying bowel or crucial motor nerves in the brachial plexus or pelvic regions for other tumor types [47,48,51,52,53]. For breast cancer, US guidance only visualizes the ice ball as it progresses beyond the anterior tumor margin as it approaches the skin. Lateral viewing by US provides limited visualization of the posterior aspect of the ice ball [43,52]. CT guidance thus allows for identification of the posterior margin of the ice which is more susceptible to greater warming by the underlying natural body heat (Figure 2b). Once the freeze has begun and the probes are thoroughly stuck to the progressing ice, CT guidance also allows for the monitoring of the anterior retraction of the probes to physically lift the posteriorly developing ice off the pectoralis muscle. Caution should be noted during this anterior retraction by more careful monitoring of the anterior ice margin as it may quickly approach the skin surface [43].

Hydrodissection with 0.9% saline is used for safety in nearly every case where the anterior tumor margin approaches within 5 mm of the skin surface. Figure 4 shows the 2 components of the “knuckle rule”, whereby close tumor proximity can be evaluated for potential skin involvement. The ability of the skin to flexibly accommodate the injected saline to significantly increase the safety margin of the underlying ice progression can also be assessed. As long as a subcutaneous breast cancer does not produce non-movable, fixed skin over its palpable aspect, sufficient hydrodissection usually involves between 10 and 50 cc for most smaller tumors (<2 cm) near the skin (<5 mm). Hydrodissection can also be used posteriorly to better displace the breast cancer target from the underlying pectoralis muscle, thereby avoiding some of the soreness associated with healing. However, freezing that extends into the pectoralis muscle and chest wall is generally safe and well-tolerated. As described in the following section, extensive hydrodissection using approximately 250 cc of fluid can effectively protect a large portion of the breast.

### 3.3. The Potential of Large-Volume Breast Cryoablation

We have performed several outpatient large-volume breast cryoablations in selected patients with bulky disease who declined standard treatment. Results to date have shown exceedingly low complications and side effects from large-volume soft tissue ablation and produced minimal pain [43,47,48,51,52,53]. As a palliative procedure, we hoped to prevent the debilitating pain from potential chest wall invasion and/or avoid the difficult management of direct skin involvement, with chronically draining open neoplastic wounds. In this review, we briefly describe a large-volume breast cryoablation procedure which demonstrated that it is feasible and safe as an outpatient procedure (Figure 5). However, cryoablation for T1c-3 tumors (1–>5 cm) requires further evaluation of these difficult-to-treat patients.

## 4. Immunotherapy

### 4.1. Systemic Alone (Currently Intravenous)

The primary goal of immunotherapy is to enable the immune system to recognize cancer cells as abnormal and mount an effective inflammatory response against them. In the context of breast cancer, previous reviews have examined the complex interplay between the immune system and tumor biology [21,58,59,60]. These immunotherapeutic approaches in oncology include the use of cancer vaccines that promote immune recognition of tumor-associated-antigens, the intratumoral injection of oncolytic viruses to generate localized immune response, adoptive cell therapies such as chimeric antigen receptor (CAR) T cells, and monoclonal antibodies targeting immune checkpoint treatments [21,58,59,60]. This review will briefly highlight the role of antibodies in targeting immune checkpoint pathways such as PD-1, PD-L1, and CTLA-4, which function as regulatory mechanisms that limit T cell activation and tumor-specific immune responses.

Immune checkpoint inhibition (CPI) trials with other cancers have mainly focused on antibody-targeting programmed cell death 1 protein (PD-1), programmed cell death ligand 1 (PD-L1) and cytotoxic T-lymphocyte antigen-4 (CTLA-4) [21,58,59,60]. While these trials have only shown modest success and relatively mild immune responses, combining these checkpoint inhibitors also produces severe adverse events, acknowledged to occur in 55–95% of patients [60]. Systemic immunotherapies using CPIs (i.e., intravenous administration) have revolutionized much of cancer care but also present a significant opportunity for the de-escalation of the severity, incidence and cost of these side effects.

The use of any CPIs for breast cancer has had limited response rates of 0–16%, leading to the perception that breast cancers are “cold” to immunotherapies. Drug combinations have been primarily limited to pembrolizumab or atezolizumab and/or combined with standard chemotherapy drugs of paclitaxel, carboplatin and gemcitabine [21,58,59,60]. Most patients in these trials experienced the lower complication rates noted for individual drug administration, rather than the higher rates of adverse events when anti-CTLA-4 drugs are combined with PD-1 (PD-L1) inhibitors. More recent reviews of systemic immunotherapy for breast cancer have mainly focused on adaptive cell therapies using innate immune cells, such as dendritic cells, natural killer cells and chimeric antigen receptor T cells (CAR-T) [21]. Such cell-based therapies are generally very expensive and have limited insurance coverage currently, despite some Food and Drug Administration (FDA) approvals. The complexity and expense of current systemic immunotherapies for breast cancer thus presents a significant opportunity to reduce metastatic cancer care costs with alternative approaches that could de-escalate serious complication rates while reducing costs from combined intravenous CPIs.

Overcoming the relative resistance of breast cancer to current immunotherapies may require further combination with strategies that enhance tumor immunogenicity, such as ablations that release a panoply of membrane antigens that do not need to be individually targeted. The merits of each form of clinically available ablation techniques have been reviewed, but heat-based ablations (e.g., radiofrequency, microwave) tend to denature proteins. Conversely, cryoablation has shown the preservation of the collagenous architecture of ablated tissues [43,48,51,52,53], as well as the integrity of surface membrane antigens, to cause a local inflammatory response that also releases cellular stress signals and type I cytokines. These tend to recruit antigen-presenting cells to the tumor and induce a tumor-specific T cell response and markedly elevated pro-inflammatory cytokines, especially when compared to heat-based approaches [61,62,63]. If local cryoablation carries minimal procedural risks and side effects, why should we tolerate the high morbidity associated with intravenous administration of checkpoint inhibitors (CPIs), when direct intratumoral injections could potentially deliver higher tissue concentrations, enhance efficacy, and reduce both systemic side effects and overall costs?

### 4.2. Human IntraTumoral ImmunoTherapy (HIT-IT; Drugs Alone)

Growing evidence supports intratumoral immunotherapy (HIT-IT) as a potent strategy to prevent regional and metastatic progression in solid tumors, including breast cancer. Extensive work has been carried out over the last decade exploring the future role of exchanging intravenous administration of chemotherapeutic agents with the intratumoral approach which delivers much higher local drug concentrations [61,62,63,64,65,66,67,68,69,70,71]. Marabelle et al.’s [66,67] pioneering work provided strong evidence showing that initiating an immune response directly within the tumor microenvironment can induce both localized cytotoxic effects and systemic immune responses (abscopal effect), leading to the regression of untreated lesions beyond the injection site. This prompted an international conference during which key tumor classifications and parameters were established to evaluate and standardize intratumoral therapies. Given the observation of abscopal responses, it became crucial to distinguish between the injected (enestic) lesions and the non-injected (anenestic) lesions (Figure 6), with treatment responses monitored using the established iRECIST criteria [69]. This includes technique considerations that also apply to future ablation immunotherapy approaches, such as using a multi-hole needle for more symmetric injection of the drugs [70].

Overall, extensive work by Xu et al. [64], Yuan et al. [65], and Champiat et al. [66], among others, have validated the beneficial role of follow-up tissue biopsies and tailoring intratumoral injections to specific tumor regions, particularly those identified on PET/CT scans as metabolically active (i.e., elevated standardized uptake values SUV). These studies have shown that such biopsies can pinpoint immunologically active tumor subregions and help address the inherent heterogeneity found within solid tumors [69,70,71]. The importance of defining the safety profile of interstitial injections across various agents was highlighted in multiple studies, including those of Ghosn et al. [67] and Munoz et al. [70], which also demonstrated the advantage of achieving higher local tissue concentrations with significantly lower intratumoral doses compared to intravenous administration. In addition, Sheth et al. provide real-world insights into the challenges and feasibility of image-guided intratumoral delivery across different tumor histologies and anatomical sites, reinforcing the practicality of this approach [68].

Guidelines for evaluating tumor response to immunotherapy have evolved since Seymour et al.’s [69] introduction of iRECIST criteria to account for both direct (enestic) and systemic (anenestic) effects. Munoz et al. [70] and den Brok et al. [71,72] further supported these developments by emphasizing the significance of monitoring off-target responses and the immunologic implications of localized interventions. Additionally, Mauda-Havakuk et al., in a preclinical colon cancer model, demonstrated that antigen release following cryoablation or RFA enhance systemic immune activation, thereby, promoting abscopal effects in non-targeted tumor sites. The study provided valuable information on immune responses to ablation techniques [73]. In summary, these collective studies underscore the value of HIT-IT in increasing local the bioavailability of immunotherapeutics while minimizing systemic exposure, reducing associated toxicities, and potentially lowering overall treatment costs. The minimally invasive nature of image-guided injections and biopsies offers a compelling alternative to traditional intravenous checkpoint inhibitor regimens, as noted across studies [64,65,66,67,68]. It remains critical to emphasize that the efficacy of HIT-IT in breast cancer specifically has yet to be formally established, highlighting the need for further IRB-approved clinical trials in this area.

### 4.3. The Potential of Activating Immune Responses

#### Combining Cryoablation + Immune Stimulating Adjuvants

Cryoablation and its immunological effects have been well documented in the literature [61,62,63], but it is worth noting that the documented differences between heat-based ablations and associated immune adjuvants now date back almost 20 years [72,74,75,76]. Heat-based ablations provide some immune stimulation alone as evidenced in the antigen loading of dendritic cells (DCs), which is a prerequisite for the initiating adaptive immune responses. More recent assessments of the tumor microenvironment also confirm limited immune stimulation from radiofrequency ablation. In contrast, cryoablation stimulates a broader secretion of cytokines, suggesting a more robust immunogenic response [76]. CpG oligodeoxynucleotides ODNs are synthetic DNA molecules that are commonly used as adjuvants to enhance vaccines. These agents are used to improve the immunogenicity of vaccines and can be delivered systemically or mucosally. CpG ODNs are particularly effective in boosting immune responses in groups with reduced immune function, such as the elderly and immunosuppressed. While commercially available toll-like receptor 9 (TLR9) adjuvants (e.g., Cytosine-phosphate-Guanine; CpG) are not yet approved for intratumoral injection in humans, preclinical studies consistently demonstrate their benefit in stimulating the immune system [77].

Consequently, cryoablation has undergone the most comprehensive assessments regarding its induction of anti-tumor immunity [78]. In a study by McArthur et al., the combination of intravenous checkpoint inhibitors and cryoablation tested in a small cohort group of human breast cancer patients led to increased expression of T cell activation markers, elevated serum Th1 cytokine levels and a reduction in immunosuppressive CD4^+^PD-1^hi^ T cells, resulting in an improved effector-to-suppressor T cell ratio. These findings indicate that this combined approach is not only safe but also holds promise for generating a synergistic antitumor immune response [79].

## 5. Combined Cryoablation and HIT-IT-Type Ablation Immunotherapy (Current and Future)

Finally, we present initial findings from the first known clinical trial combining cryoablation with intratumoral immunotherapy (NCT04713371), which demonstrated the feasibility of inducing targeted tumor regression, even with subtotal cryoablation, and observed distant abscopal effects (Rampart Health, Orlando, FL, USA) [80]. The Phase 2 clinical study known as the Abscopal 5001 trial evaluated the safety and therapeutic potential of the Multiplex Combination Intratumoral Immunotherapy (MCII) in patients with metastatic solid tumors. The protocol involved image-guided cryoablation in combination with localized injections of low-does immune checkpoint inhibitors and chemotherapy agents. Thirteen participants, including twelve with metastatic cancers and one with sacral chordoma, received at least one cycle of intratumoral MCII following a 3–5 course of oral cyclophosphamide intended to reduce T-suppressor cells. The treatment involved CT-guided cryoablation, followed by the injection of approximately 2 cc each of ipilimumab, pembrolizumab, and cyclophosphamide directly into the tumor. To prevent leakage and ensure tract hemostasis, the needle path was sealed with about 4 cc of Helitene (Integra Lifesciences, Princeton, NJ, USA) upon withdrawal. Granulocyte-macrophage colony-stimulating factor (GM-CSF) was administered subcutaneously over four weeks to enhance immune activation. Repeat treatment cycles were allowed at six-week intervals if imaging, assessed by iRECIST criteria, showed stable disease or tumor reduction [80].

Figure 7 demonstrates cryoimmunotherapy targeting lung and axillary tumors. Pre-MCII treatment CT images (A–C) show two pulmonary lesions, one in the right upper lobe and another in the perihilar region in a 68 year old patient with metastatic sarcoma for which his right distal arm had been amputated. The most prominent metastasis was measured up to 3.6 cm in the right axilla with an additional 1.5 cm subcutaneous focus in the remaining arm stump. The few pulmonary metastases were noted, as well as a left rib mass which had undergone radiation therapy for local control and pain relief.

Middle row of Figure 7D–F shows the ultrasound-guided procedure a parallel set of 18-gauge needles, placed ~7 mm from a central intratumoral 1.7 mm short segment freeze cryoprobe ((IceSeed; Galil Medical, Yokneam, Israel), (BostonScientific, Marlborough, MA, USA) on the first MCII session (Figure 7D–F). Six months after the first MCII procedure, the two enestic lung lesions had nearly completely regressed (Figure 7A*–C*). The right axillary targeted mass had regressed 59% by iRECIST criteria, leaving the primary tumor only 2.5 × 1.5 cm at the time of the second treatment. Moreover, repeat biopsy showed only necrotic debris and no viable tumor. These findings corroborate the main outcome of the limited trial of clear abscopal response and primary tumor target reduction with necrosis [80].

The cohort included four cases of prostate cancer, two of sarcoma, and one case each of breast, colon, bladder, cervical, tongue, kidney cancer, and sacral chordoma. Eight participants underwent at least three treatment cycles, while two received two cycles, and three completed only one. The procedure was well-tolerated across all patients, with each discharged within two hours of the outpatient intervention. The overall adverse event rate was 69%, with nearly all events classified as Grade 1 or 2, aside from a single delayed cryoablation-related complication. Most side effects were minor, such as temporary skin rashes commonly associated with subcutaneous GM-CSF administration. Localized injection site reactions were recorded in nine patients (69%), and systemic abscopal responses were observed in four individuals (31%), including one sarcoma patient who experienced the complete regression of pulmonary metastases (Figure 7). Patients are being followed to assess the long-term effectiveness of the treatment [80].

Beyond the 31% observed abscopal effect for MCII, the encouraging 69% reduction in the targeted masses suggests that local tumor immune effects are also likely controlling the local recurrence rate despite subtotal partial necrosis by cryoablation [80,81]. Indeed, if cryoablation alone had been performed to partially treat these masses, a brisk recurrence of adjacent viable tumors could have happened, perhaps due to the stimulation of growth from the associated neovascularity of the healing ablation rim. This is similar to the lack of brisk progression noted with some HIT-IT chemo/immunotherapy injections alone. Intratumoral immune injections adjacent to a limited cryoablation site offer opportunities for the de-escalation of a broad spectrum of current treatments for breast cancer, especially if procedural costs are reduced as much as the minimal adverse events.

## 6. Cost-Effectiveness and Patient-Reported Quality-of-Life Outcomes

Cancer patients are likely to experience financial hardship associated with the disease [82,83,84,85]. In 2020, the national cost of breast cancer was estimated at USD 29.8 billion [82]. The concept of de-escalation applies not only to the potential reduction in morbidity, but may significantly reduce economic costs that drive more favorable decisions for insurance coverage of not only breast cryoablation alone, but also future intratumoral ablation immunotherapy. Cryoablation is now a minimally invasive treatment for breast tumors that offers several advantages; including minimal pain, shorter recovery periods, better cosmetic outcomes and cost-effectiveness when compared to traditional surgery, and other costly treatments [46,86,87,88,89,90,91,92].

When comparing cost-effectiveness and patient-reported quality of life for breast cryoablation alone, it was concluded that cryoablation has an 87% lower cost than surgical resection (USD 2221.50 for outpatient cryoablation vs. USD 16,896.50 for surgical resection with general anesthesia) [46]. In addition, the Breast Q survey, administered a year post-cryoablation, revealed notably better patient satisfaction with physical breast appearance, sexuality, and improved financial well-being than those with surgical resection or lumpectomy [46]. However, their calculation of direct technical costs for outpatient cryoablation of USD 2221.50 is probably too low for most institutions to provide as an out-of-pocket patient charge, especially since each cryoprobe costs ~USD 1500. Yet, many institutions currently offering breast cryoablation have projected technical components for breast cryoablation up to USD 30,000 per procedure, which may be exorbitant and potentially seen as opportunistic.

We therefore present a reasonable estimate of de-escalation opportunities for the current major treatment options for all stages of breast cancer. This obviously cannot be carried out with individual patient treatment estimates since costs are highly variable, depending on treatment stages. We therefore made a “best-guess” at quantifying costs for the major treatment modalities noted in Table 2 [82,83,84,85,86,87,88,89,90,91,92,93,94,95]. Table 2 shows estimated costs of current breast cancer treatments. The largest opportunities for de-escalation of costs and treatment morbidity may be for metastatic disease, currently using chemotherapy and intravenous immunotherapy. Recent enthusiasm for cryoablation may be more for the de-escalation of treatment morbidity than marked cost savings for early breast cancers. Therefore, the possibility of percutaneous ablation (i.e., cryoablation) in combination with intratumoral injection of immunotherapy (and possibly chemotherapy) may produce the greatest de-escalation of both costs and the severe systemic side effects of immunotherapy and/or chemotherapy.

The most cost-effective treatment in the table appears to be for adjunctive hormonal therapy at approximately USD 200/year, which also had relatively low severe complication rates as noted in Section 2. Surprisingly, intravenous immunotherapy had the highest costs, especially when combining checkpoint inhibitors (CPIs) for at least USD 400,000/year. Notably, intratumoral immunotherapy using both CPIs includes a sufficient volume in a single vial to support three separate intratumoral treatment sessions, using approximately 2 cc per session. The total cost of the primary drugs used across all three session is 19,527 (i.e., USD 6098 + USD 9174+ USD 4255), which is over 20-fold less expensive than a full intravenous course of Keytruda + Yervoy. Moreover, the combined rate of severe complications associated with IV-administered CPIs exceeds 50%, as noted in Section 4.1 [60]. Given that CPIs have revolutionized immunotherapy and have been given regulatory approval with an associated high price, it appears to be more imperative that more expanded intratumoral immunotherapy trials are established. Advancing the use of intratumoral therapies will require greater involvement from pharmaceutical companies to sponsor and support clinical trials to demonstrate efficacy comparable to intravenous (IV) regimens [92,93,94,95]. Table 2 supports preliminary cost comparisons between breast cryoablation and surgical resection [46]; however, potential savings are effectively still less than ~USD 15,000, despite surgery being 5–7 times more expensive. Similarly, if radiation therapy could be avoided, the maximum potential savings would exceed ~USD 10,000. Chemotherapy regimens are highly variable and are continuing to evolve; nevertheless, assuming a median price of USD 55,000, chemotherapy costs are easily double those of surgery or radiation therapy. As with immunotherapy, the higher costs of chemotherapy are also associated with a relatively high incidence of serious adverse events, which can limit patients’ tolerance and quality of life. These effects stand in stark contrast to the minimal side effects of percutaneous ablations, even when combined with intratumoral chemotherapy and/or immunotherapeutics. Demonstrating comparable efficacy between intratumoral and intravenous immunotherapy will likely depend on expanded clinical trials. In this context, collaboration with pharmaceutical companies may be important to help facilitate and fund such studies, particularly as interest grows in localized, cost-effective treatment strategies.

## 7. Discussion/Future Directions

In summary, the current standard of care for breast cancer involves surgical resection followed by hormonal therapy, chemotherapy, and/or radiation therapy to reduce the risk of recurrence and prevent metastatic progression. Each of these treatments come with an associated range of risks, complications and financial burdens. The recent literature has proposed that treatment “de-escalation” with cryoablation should be limited to tumors < 1.5–2.0 cm in elderly patients and/or those with significant comorbidities that make them a greater surgical risk [2,3,4,5,6,7,8,9,10,11]. However, our data suggest that cryoablation is not inherently limited by tumor size and may be effectively applied to a broader range of tumors currently treated with surgical resection. Limiting ablation to only the smallest breast tumors may inadvertently preserve dependence on systemic adjuvant therapies intended to address the risk of nodal involvement and distant metastases.

We therefore structured this comprehensive review to briefly address the complications associated with current standard breast cancer treatments and to highlight opportunities for treatment de-escalation across the disease spectrum. We emphasized the primary driving force of image-guided percutaneous treatment showing minimal additional morbidities for both intratumoral injections of chemo- and/or immunotherapeutic drugs, as well as ablation strategies. The reduction in adverse events by using intratumoral ablation immunotherapy also carries the potential for significant cost savings, provided that these promising approaches are more thoroughly explored. In the future, cost-effective, low-morbidity breast cancer treatments may improve patient care and quality of life by reducing reliance on systemically administered checkpoint inhibitors and their associated intravenous toxicities of CPIs. Some have suggested that these techniques are “not ready for prime time”, yet they already exist and await broader adoption and appropriate insurance coverage. This situation is similar to the technical and economic challenges previously faced during the transition from traditional exploratory surgery to image-guided tumor biopsies. Specific to breast cancer, we have already shifted from open surgical biopsies by hook-wire localization and surgical excision to minimally invasive needle biopsies of suspicious mammographic, ultrasound or MRI-guided findings. These advances have led to substantial improvements in patient quality of life and reductions in cost, strategies that now appear to be ready for broader application through intratumoral therapies and image-guided interventions.

Percutaneous image-guided access to intratumoral drug administration has been thoroughly investigated [61,62,63,64,65,66,67,68,69,70] and based on available evidence, we covered known drug combinations and ablation sources that provide optimal immune stimulation. Cryoablation alone has the longest practical history of observing the abscopal effect when used in combination with additional adjuvants and immunotherapeutic drugs [71,72,73,74,75,76,77,78] but was initially observed with radiation therapy alone [31,32]. For near future exploration, recent data suggest that cryoablation’s immune-stimulatory superiority over heat-based treatments may be outperformed by non-thermal tumor disruption methods such as pulsed electric field (PEF) ablation [81,96,97,98,99,100,101] and histotripsy [102,103]. A key practical distinction lies in commercial readiness: cryoablation systems are more widely available from multiple manufacturers, while PEF and histotripsy technologies are still undergoing clinical approval and are currently limited to single-device manufacturers.

A near-future human trial could be implemented to include direct comparisons of the immunologic outcomes between cryoablation, pulsed electric field (PEF) ablation and histotripsy particularly when paired with intratumoral immunotherapeutic regimens (e.g., checkpoint inhibitors, cytokines, or chemotherapeutic agents). This review hopes to accelerate the implementation of such trials by highlighting the very low morbidity and reduced cost of intratumoral injections, as well as introducing optimal ablation sources.

Larger clinical trials are promptly needed to better identify the immune-protective abilities of these ablation immunotherapy combinations, especially in relation to tumor size-related risks of nodal and distant metastatic involvement. Namely, the eventual dream of transforming each patient’s tumor into a personalized vaccine against additional tumor spread is highly enticing but fits the concern of “not ready for prime time use”. Critical questions remain, such as identifying the most effective intratumoral drug combinations, especially in relation to tumor size and its association with nodal and distant metastasis. Perhaps more importantly, we also need to understand the duration of both abscopal and local tumor control responses. Encouragingly, the safety of intratumoral delivery has already been demonstrated. This represents a promising opportunity to harness the innate capabilities of the immune system guiding safe and effective interventions across multiple stages of breast cancer treatment.

While we do not wish to overstate the potential of immunotherapy, intratumoral administration represents a promising advancement in the marked reduction in the combined adverse events of the intravenous administration of checkpoint inhibitors, let alone their costs [60]. For patients with significant comorbidities, or those who decline standard therapies, the addition of intratumoral immunotherapy as an adjunct to larger-volume ablation may offer a means to prevent local recurrence and potentially reduce nodal or distant metastatic progression. These patients represent a clinically relevant group and may be safely included in long-term follow-up studies assessing the outcomes of ablation-immunotherapy combinations. Similarly, the encouraging ~70% local target tumor response rate observed in ongoing studies [80,101] offers promising opportunities to reduce tumor burden in cases involving the skin and/or pectoralis muscle. These forms of extensive local invasion, which are traditionally difficult to treat, may be significantly diminished through the use of intratumoral ablation immunotherapy. In select cases, this approach may reduce tumor volume enough to enable completion with cryoablation alone, potentially eliminating the need for more extensive or disfiguring surgical procedures.

As clinical experience with intratumoral strategies continues to grow, so too does the potential for more personalized, less toxic, and more accessible cancer therapies. These approaches offer a path toward improved local control, enhanced patient quality of life, and expanded access to curative-intent treatment options, especially for patients previously considered to be ineligible for standard care. With continued research, collaboration, and support, these innovative therapies may help reshape the future of breast cancer treatment.

## 8. Conclusions

Breast cancer has long presented a management challenge for each stage of disease, often requiring combinations of surgery, radiation and systemic therapies including hormone, chemotherapy, and immunotherapy combinations to prevent eventual lethal systemic spread. Unfortunately, these standard treatments are frequently associated with significant adverse events with a relatively low chance of a “cure”. We have now entered an era wherein image-guided treatments offer the possibility of substantially lower complication rates and the potential for much lower costs compared to current standard treatments that patients have had to accept and just “battle through”. Intratumoral injection of immunotherapeutics, such as checkpoint inhibitors and cytokines, have demonstrated significantly lower complication rates compared to traditional intravenous delivery. Moreover, patients undergoing image-guided ablation procedures are often discharged within hours and experience a minimal recovery time.

Both intratumoral injections and percutaneous ablations have shown consistent safety, and little remains to prevent further clinical exploration of their combined application. Any anti-tumor benefit derived from ablation immunotherapy is further enhanced by the absence of significant morbidity, offering a clear path towards meaningful treatment de-escalation for a broader population of breast cancer patients.

Critical Relevance Statement: Cryoablation has been used for a spectrum of breast cancer treatments, and remains capable of delivering thorough ablation outcomes, regardless of tumor size. However, the persistent challenge of nodal and distant metastasis underscores the limitations of localized treatment alone. Emerging evidence supports the critical role of combining cryoablation with the intratumoral delivery of immune-modulating agents to not only maintain local control but also stimulate systemic anti-tumor immunity. This integrated approach offers a compelling opportunity to reduce treatment-related morbidity and cost, while potentially addressing metastatic spread, an area where current therapies fall short. Immediate clinical applicability and the need for protocol optimization make this a priority area for translational research.

Key Points:

•Combined CT/US, image-guided cryoablation displays visible ice extending beyond tumor margins for improved local tumor control in patients with breast cancers > 1.5 cm.•Appropriate cytotoxic isotherms need to volumetrically extend beyond all apparent tumor margins, which favors multi-probe cryoablation for tumors > 1.5 cm, whereas even larger single cryoprobes appear to be limited to breast cancers < 1.5 cm.•Intratumoral injection of chemo- and immunotherapeutic drugs is a promising strategy for improving the effectiveness and reducing the morbidity of chemotherapy and immunotherapy.•Image-guided cryoablation combined with intratumoral immunochemotherapy provides a dual-modality approach that promotes systemic immune responses and may markedly decrease costs of future breast cancer therapies.

## Figures and Tables

**Figure 1 cancers-17-01915-f001:**
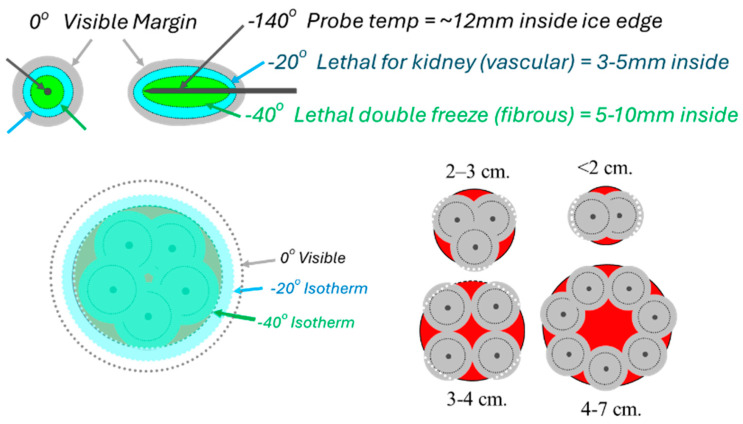
The 1–2 rule for cryoprobe placements according to tumor size, whereby standard cryoprobes (2.1/2.4 mm diameter) are placed no farther than 1 cm from the tumor margin and <2 cm apart. If a single probe is used for very small masses, a precise central location is needed to cover the tumor with ice (*outer gray circles around each probe*), and lethal ice (*smaller dashed line* ~*5 mm inside ice*). Conversely, when more probes are used, the sum of the individual ice projections (*darker gray circles*) synergistically expand, extending lethal ice (*larger dashed line*) ~5 mm beyond all tumor margins while visible ice (*lighter gray circle*) projects ~10 mm.

**Figure 2 cancers-17-01915-f002:**
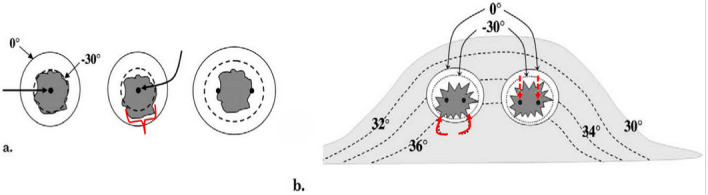
Ref. [43] (**a**) Diagram shows basic isotherms for single (**left and center**) and double cryoprobes (**right**). Accurate central placement of a single 2.4 mm cryoprobe (left image, larger black arrow) within a simulated 1.5 × 1.5 cm tumor (dark gray) may still not produce sufficient lethal ice to cover all tumor margins (dashed line ~<30 °C, diameter ~1.5 cm). Even though visible ice (solid outer circle) may appear to cover all tumor margins, slight off-center placement (**middle** image, curved black arrow) leaves a grossly untreated tumor (**red** bracket) beyond the lethal isotherm (dashed circle). Tumor on the right is covered by lethal ice due to synergy produced by two cryoprobes [47]. (**b**) *Avoiding posterior positive margins**:* heat load effects of the chest wall. The estimated temperature difference between the skin surface (30 °C) and chest wall/body (36 °C) causes greater heat load along the posterior margin of ice propagation, which narrows the posterior distance between the visible (0 °C) and lethal (–30 °C) isotherms (curved solid arrows). Ablation on the left shows the central position of cryoprobes and greater anterior extension of visible ice beyond the tumor margin; however, incomplete coverage of posterior tumor margins (**red** curved arrows) is noted, like that seen in prior series [40]. Ablation on the right shows thorough tumor coverage by lethal ice due to the more posterior placement of cryoprobes in the tumor (**red** straight arrows), thus overcoming the heat-sink effect along the chest wall [43].

**Figure 3 cancers-17-01915-f003:**
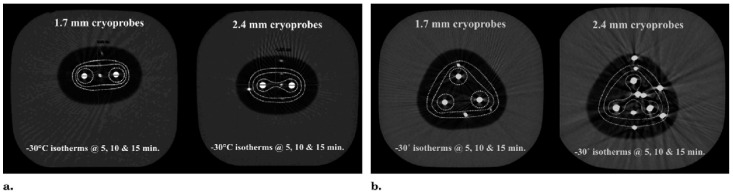
Ref. [47]. Progression of the lethal isotherm (−30 °C) at 5 min (inner dashed white lines), 10 min (dashed intermediate lines), and 15 min (dashed outer lines) is shown for double and triple JT Ar-based 1.7 and 2.4 mm cryoprobes (larger central solid circles), overlaid on the CT image for the total ice appearance at 15 min. (**a**) With a double cryoprobe (largest 2 central bright circles) configuration, the lethal ice surface area grows more for 2.4 mm cryoprobes (**right**) after 5 min as a result of early synergy than for 1.7 mm cryoprobes (**left**). Smaller peripheral bright solid circles represent thermocouples documenting the isotherm temperatures. (**b**) With a triple-cryoprobe configuration, the lethal ice surface area grows more for 2.4 mm cryoprobes (**right**) after 5 minutes as a result of early synergy, but the difference becomes less over time after synergy also occurs for 1.7 mm cryoprobes (**left**). Smaller bright peripheral circles are again the temperature-validating thermocouples.

**Figure 4 cancers-17-01915-f004:**
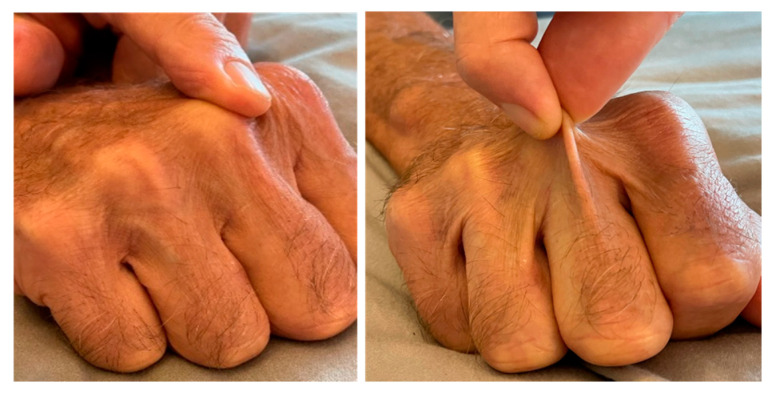
Hydrodissection planning and two components of the Knuckle Rule. For tumors <~5 mm from the surface, mobility should feel like the skin sliding over a knuckle (**left**) to allow the injected fluid to thicken the skin by >2 mm. If you can pick up the skin even a little (**right**), then you can thicken the overlying skin > 10 mm which is especially needed for large-volume ablations.

**Figure 5 cancers-17-01915-f005:**
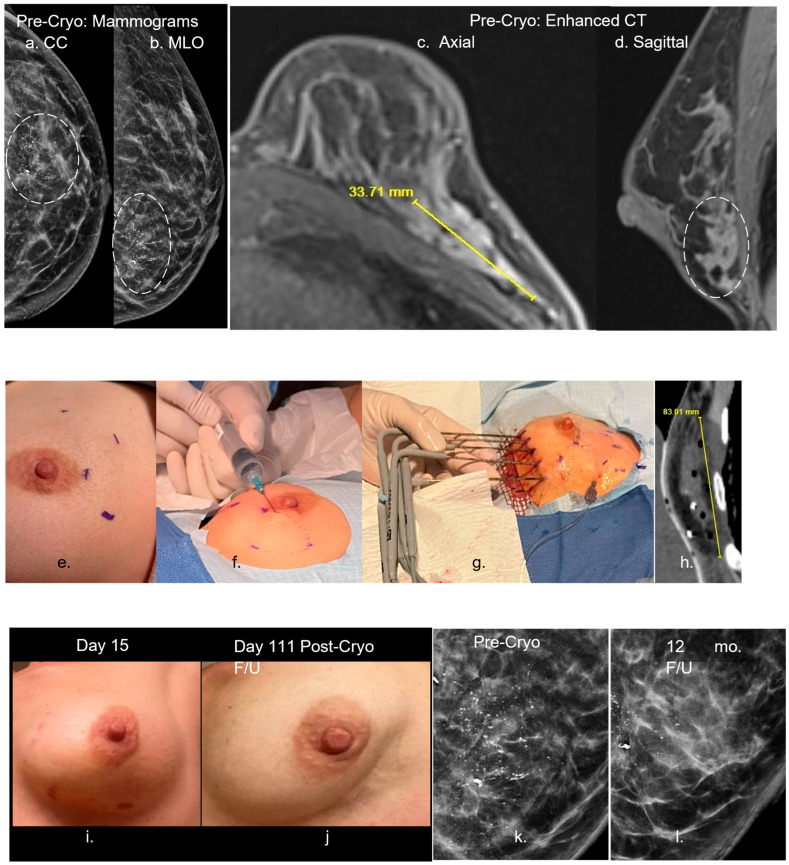
Ref. [43] (**a**–**d**) Pre-cryoablation mammograms (**a**,**b**)-left (dashed circle)) show a large grouping of malignant calcifications, compatible with the CT enhancement (**c,d**) extending up to 3.4 cm in the lower lateral left breast on axial (**c**) and sagittal (**d**) images. Procedural images (**e**–**h**) show initial planning marks on the skin (**e**) with thorough hydrodissection over the lateral aspect of the enhancing tumor (**f**). Five cryoprobes were placed medially (**g**) and extended throughout the interior breast as noted on the immediate postprocedural CT (**h**) showing a low-density ice coverage of ~8 cm craniocaudal. (**i**–**l**) The two post-procedure left images (**i**,**j**) show minimal residual bruising at day 15, with eventual resorption and mild overlying skin retraction of the left lower lateral breast by day 111 (~4 mo). The two right magnified mammograms compare the pre-cryoablation and 12-month follow-up appearance of the near complete resorption of the diffuse region of calcifications. The residual posterior calcifications appeared to be relatively stable and may represent residual benign calcifications but further long-term follow-up is needed to assess whether malignant calcifications resorb after cryoablation more so than benign calcifications. Moreover, no significant coarse calcifications of fat necrosis were noted. The described large volume breast cryoablation procedure required sufficient procedural experience and/or image guidance skills with US and CT, following careful consideration of the breast MR “roadmap”, and associated three-dimensional CT to plan for thorough coverage. However, our objective of ablating nearly any size breast tumor is also similar to multi-probe US/CT-guided procedures in nearly any other organ. For instance, Wang et al. found successful local control across a wide range of thoracic tumor sizes, highlighting the adaptability of cryoablation beyond strict size constraints. Similarly, Littrup et al. demonstrated that renal tumors of various sizes, including those >3 cm, were effectively managed through percutaneous cryoablation guided by CT. Moreover, in their long-term study of hepatic tumors, they showed that cryoablation provided consistent tumor control across a diverse patient cohort, including those with larger and anatomically complex lesions [48,51,52,53]. Holmes and Iyengar maintain that although cryoablation technology is optimized in ultrasound-visible stage 1 breast cancer, technique modifications can permit the cryoablation of stage 0, II, III and IV. The authors reveal that cryoablation achieves durable local tumor control in appropriately selected breast cancer patients with ongoing trials supporting its use as a definitive treatment modality in select populations. The authors emphasize that when combined with multidisciplinary oversight, cryoablation can serve as a viable alternative to surgery, offering effective control with minimal morbidity [10]. In summary, cryoablation of nearly any size breast cancer appears to be feasible, yet also may still require adjunctive treatments similar to surgical resection in order to manage the greater likelihood of nodal and distant metastatic disease from larger breast cancers. While current adjunctive treatments—such as hormonal therapy, radiation, and chemotherapy—are essential for managing regional and metastatic breast cancer, they are often associated with substantial side effects and morbidity. Similarly, intravenous immunotherapy carries a high risk of adverse effects. The likelihood of breast cancer recurrence after cryoablation is closely tied to ensuring complete cytotoxic coverage that extends beyond all visible tumor boundaries. Because of this, precise visualization of tumor margins may be even more critical in cryoablation than in surgical resection. Any residual viable tumor left behind may be stimulated by the surrounding hypervascular healing zone, potentially leading to rapid regrowth. Although this risk also exists with incomplete surgical excision, residual tumor proliferation following cryoablation can occur more aggressively, necessitating prompt re-treatment. The remaining sections in this review also offer hope that these recurrence risks, side effects and morbidities may also be diminished when cryoablation is combined with intratumoral immunotherapy. The dashed circles in figures (**a**,**b**,**d**) is the grouping of malignant calcifications.

**Figure 6 cancers-17-01915-f006:**
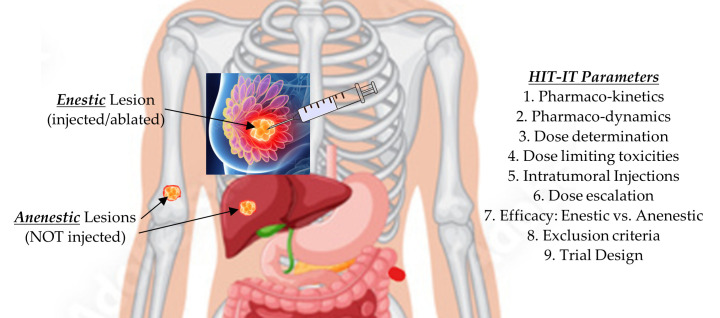
Points to consider when designing an intratumoral immunotherapy clinical trial [64,65].

**Figure 7 cancers-17-01915-f007:**
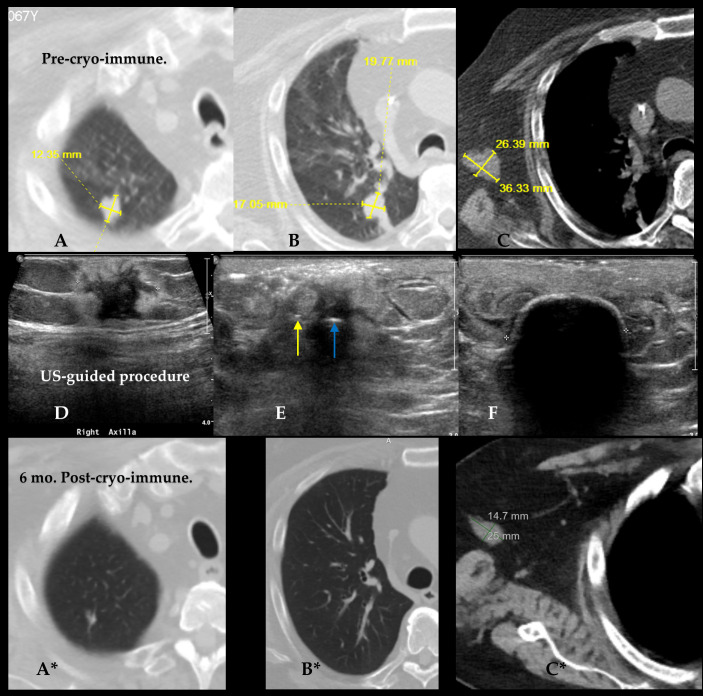
Top (**A**–**C**) and bottom (**A***–**C***) row of images show 3 pre- and post-ablation immunotherapy comparisons across anatomic areas. In Pre-Cryo-Immunotherapy CT images, the 2 lung lesions are visualized in the right upper lobe (**A**) and right perihilar region (**B**), each measuring up to ~2 cm. Axillary lymph node metastasis is shown in (**C**), measuring up to 3.6 cm in maximum diameter. Center row (**D**–**F**) display the ultrasound-guided cryoablation mid-procedure images confirming accurate cryoprobe insertion and iceball formation. Arrows in (**E**) show the single cryoprobe (blue arrow) and injection needle in (yellow) positioned ~7 mm from the cryoprobe. Image (**F)** (middle row, right) shows the 2 cm diameter echogenic ice rim with shadowing nearly covering all aspects of the tumor at 4-min freeze.  Bottom row of CT images (**A***–**C***) obtained 6-months post-cryo-immunotherapy show the same anatomic sites with substantial tumor reduction. Right lung lesions (**A***, **B***) show radiological resolution with no visible mass. In image (**C***) axillary nodes shrinks to 14.7 × 25 mm, reflecting significant immunologic and ablative response.

**Table 1 cancers-17-01915-t001:** Tumor size (pT) and corresponding TNM staging, treatments and metastasis.

pT	pTNM-Stage	Clinical Treatments	Metastasis
**Stage 0**			
Tis: DCIS, LCIS, Paget’s (no tumor)	0/Tis	Surgery, Radiation	No distant metastasis
**Stage I**			
T1 mic ≤ 0.1 cm	IA/T1N0	Surgery, Radiation	No distant metastasis
T1a ≤ 0.5 cm	IA/T1N0	Surgery, Radiation	No distant metastasis
T1b > 0.5–1.0 cm	IB/T0-1N1 or T0-1N1mi	Surgery, Radiation, Chemotherapy	No distant metastasis
**Stage II**			
T1c > 1–2 cm	IIA/T1N0-1, T2N0	Surgery, Chemotherapy, Radiation	No distant metastasis
T2 > 2–5 cm	IIA/T2N0-1	Surgery, Chemotherapy, Radiation	No distant metastasis
T3 > 5 cm (no chest wall/skin invasion)	IIB/T3N0-1	Surgery, Radiation, Chemotherapy	No distant metastasis
**Stage III**			
T4a (chest wall extension)	IIIA/T0-2N2, T3N1-2	Surgery, Radiation, Chemotherapy	Metastasis
T4b (ulceration, skin nodules, peau d’orange)	IIIA/T4N0-3, T3N1-2	Surgery, Chemotherapy	Metastasis
T4c (T4a + T4b)	IIIC/any TN3	Chemotherapy, Radiation	Metastasis
**Stage IV**			
T4d (inflammatory cancer)	IV	Systemic	Metastasis

**Table 2 cancers-17-01915-t002:** Cost comparison: de-escalation opportunities in breast cancer treatment.

De-escalation Opportunities	Estimated Costs
Cryotherapy–Small Tumor	Hospital Outpatient Facility Fee: USD 3631; Medicare pays up to USD 2905 [85]
Cryotherapy–Large Tumors (7 probes)	Outpatient Cost: USD 8226 [46]
Lumpectomy	Total Cost: USD 16,896.50 [46]; Expected Range: USD 10,000–USD 20,000 [86,88]
Mastectomy	Medicare Pays: USD 5726; Expected Range: USD 15,000–USD 55,000 [89]
Hormonal Therapy	Tamoxifen: ~USD 14/month or ~USD 168/year; Arimidex: ~USD 19/month or ~USD 228/year [90]
Chemotherapy	USD 10,000–USD 100,000 [90]
Radiation Therapy (Whole Breast)	USD 4500–USD 14,500 (specifically reported for breast cancer) [91]
Immunotherapy–Intravenous (IV)	USD 100,000–USD 500,000 (varies); Pembrolizumab: USD 22,674.72 every six weeks (~USD 191,000/year); Yervoy (3-year full regimen): ~USD 1.77 million [USD 145/mg] [92,93,94,95]
Immunotherapy–Intratumoral + Sub-Q	Keytruda: ~USD 6098 per 4 mL dose [92,94]; Yervoy: ~USD 9174 per 10 mL dose [92,93,95]; Leukine (GM-CSF): ~USD 4255 for 14 vials (4-week post-ablation course)
Cellular-Based Treatments (e.g., CAR-T)	~USD 373,000 per infusion [92]

## Data Availability

Not applicable since this review did not involve new data.

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
