# Peer review of "Cryoablation and Intratumoral Immunotherapy for Breast Cancer: A Future Path to Cost-Effective De-Escalation for Larger Tumors, Lymph Nodes and Metastatic Disease"

_cancers, 2025, doi:10.3390/cancers17121915_

Round 1
Reviewer 1 Report (Previous Reviewer 1)
Comments and Suggestions for Authors
The additional editing to the manuscript from the previous submission made the article greatly improved and much easier to follow. Only very minor technical corrections.
1. Section 3.1. Cryoablation + Adjuvants (references need corrected)
a. More recent assessments of the tumor microenvironment also confirm mild immune stimulation from radiofrequency, but cryoablation stimulates a broader secretion of cytokines [77]. While commercially available toll-like receptor 9 (TLR9) adjuvants (e.g., Cytosine-phosphate-Guanine; CpG) are not currently available for intratumoral human injection, animal studies clearly point to their benefit and have shown broad use stimulating the immune system [78-80]. References 78-80 are not correct.
b. Administration of intravenous checkpoint inhibitors plus cryoablation have also been tested in a small group of human breast cancers, which induced higher expression of T cell activation markers, higher serum Th1 cytokines and reduced immunosuppressive serum CD4+PD-1hi T cells, improving effector-to-suppressor T cell ratio [80]. Reference 80 is not correct. The manuscript referenced is an animal model of colon cancer not breast cancer. I think you meant to reference McArthur. Please refer to these 2 References.
- PMID: 27566765
- PMID: 38333710
You should double check all references throughout the manuscript to make sure they are correct. The references may be off due to adding new references and the Reference Manager not updating them correctly.
2. Figures.
a. Figure 6. Fonts in the figure are not consistent.
b. Figure 8. Labelling needs be adjusted so the letters align with the image correctly.
Author Response
The additional editing to the manuscript from the previous submission made the article greatly improved and much easier to follow. Only very minor technical corrections.
- Section 4.3.1. Combining Cryoablation + Immune Stimulating Adjuvants (references need correcting)
- More recent assessments of the tumor microenvironment also confirm mild immune stimulation from radiofrequency, but cryoablation stimulates a broader secretion of cytokines [77]. While commercially available toll-like receptor 9 (TLR9) adjuvants (e.g., Cytosine-phosphate-Guanine; CpG) are not currently available for intratumoral human injection, animal studies clearly point to their benefit and have shown broad use stimulating the immune system [78-80]. References 78-80 are not correct.
Author’s Response: Thank you for noticing the issue with the following statement in section 4.3.1 [78-80 were only partially correct]. As per reviewer #1’s suggestion. We made the corrections. Revisions to Section 4.3.1 are highlighted in red within the manuscript to indicate the specific changes made.[Updated text in the manuscript]
While commercially available toll-like receptor 9 (TLR9) adjuvants (e.g., Cytosine-phosphate-Guanine; CpG) are not currently approved for intratumoral human injection, preclinical studies consistently demonstrate their benefit in stimulating the immune system.
Additionally, we retained reference #78 Zhang, Z.; Kuo, J.C.; Yao, S.; Zhang, C.; Khan, H., & Lee, R.J. CpG Oligodeoxynucleotides for Anticancer Monotherapy from Preclinical Stages to Clinical Trials. Pharmaceutics 2021, Dec 28;14(1):73. doi: 10.3390/pharmaceutics14010073. PMID: 35056969; PMCID: PMC8780291. And we also relocated and restructured the section for references: #79 and #80 to flow with the next paragraph.
Reviewer comment:
Administration of intravenous checkpoint inhibitors plus cryoablation have also been tested in a small group of human breast cancers, which induced higher expression of T cell activation markers, higher serum Th1 cytokines and reduced immunosuppressive serum CD4+PD-1hi T cells, improving effector-to-suppressor T cell ratio [80]. Reference 80 is not correct. The manuscript referenced is an animal model of colon cancer not breast cancer. I think you meant to reference McArthur.
Author’s Response:
Thank you for noting the error in reference #80.
We have now cited the article by McArthur et al. (PMID: 27566765) as suggested. The updated paragraph reads:
Cryoablation has thus undergone the most extensive anti-tumor immunity evaluations [79]. (we are citing author Wu reference #79) here. Administration of intravenous checkpoint inhibitors plus cryoablation tested in a small group of human breast cancers, induced higher expression of T cell activation markers, higher serum Th1 cytokines and reduced immunosuppressive serum CD4+PD-1hi T cells, improving effector-to-suppressor T cell ratio [80]
Suggested reference used:
McArthur H..L.; Diab, A.; Page, D.B.; Yuan, J.; Solomon, S.B.; Sacchini, V.; Comstock, C.; Durack, J.C.; Maybody, M.; Sung, J.; et al. A Pilot Study of Preoperative Single-Dose Ipilimumab and/or Cryoablation in Women with Early-Stage Breast Cancer with Comprehensive Immune Profiling. Clin Cancer Res. 2016 Dec 1;22(23):5729-5737. doi: 10.1158/1078-0432.CCR-16-0190. Epub 2016, Aug 26. PMID: 27566765; PMCID:5161031
Reference #80 is now corrected.
Reviewer suggestions:
Figure 6. Fonts in the figure are not consistent.
Author’s Response:
Thank you for pointing out the inconsistent font. We have corrected it to ensure consistency throughout the manuscript.
Reviewer #1 says Figure 8. Labeling needs be adjusted so the letters align with the image correctly (Please note that we discovered that figure 8 is actually figure 7).
Author’s Response:
We corrected the labeling in (Figure 7) to line up with the images correctly.
Author’s concluding statement: Thank you for your valuable suggestions and assistance in correcting the reference errors. We look forward to the final publication of the article.
Reviewer 2 Report (New Reviewer)
Comments and Suggestions for Authors
This is a very interesting review. It provides a comprehensive description of cryoablation for breast cancer, and, notably, even large tumors are included. The discussion of local immunotherapy was also particularly insightful.
Author Response
Dear Reviewer 2,
Thank you for your thoughtful and encouraging review. I’m glad you found the discussion of cryoablation for breast cancer informative, particularly the inclusion of large tumors. I also appreciate your kind words regarding the section on local immunotherapy.
Reviewer 3 Report (New Reviewer)
Comments and Suggestions for Authors
The article titled "Cryoablation & Intratumoral Immunotherapy for Breast Cancer: A Future Path to Cost-effective De-escalation for Larger Tumors, Lymph Nodes, and Metastatic Disease" by Josephine Fermanian et al offers valuable insights; however, there are several areas that require attention:
- The authors demonstrated the metastatic risk in tumors >1.5 cm; however, they did not adequately address whether cryoablation alone can ensure sufficient local control in these cases. Long-term oncologic results are required.
- The authors mentioned that cryoablation can decrease the reliance on radiation/chemotherapy; however, for larger tumors, adjuvant therapy may still be essential. The claim of "thorough local control" needs stronger confirmation.
- While intratumoral immunotherapy is interesting, its efficiency in preventing regional or metastatic spread remains unconfirmed in large breast cancers. In this case, the authors should discuss more about clinical trials.
- The authors should discuss whether cryoablation works best in certain subtypes of breast cancer (e.g., ER+, HER2-).
- If cryoablation reduces the use of adjuvant therapy, does it offset the initial costs?
- My suggestion in this study is that Cryoablation’s efficacy relies on attaining cytotoxic isotherms throughout the tumor. In breast tissue, heterogeneity (e.g., dense vs. fatty tissue) may lead to varying ablation margins, increasing the risk of recurrence.
- "Cost-effective" claims are speculative—if cryoablation requires adjuvant therapies (radiation, immunotherapy) anyway, where are the savings? There is no discussion of re-treatment costs if ablation fails, leading to salvage surgery.
Author Response
The article titled "Cryoablation & Intratumoral Immunotherapy for Breast Cancer: A Future Path to Cost-effective De-escalation for Larger Tumors, Lymph Nodes, and Metastatic Disease" by Josephine Fermanian et al offers valuable insights; however, there are several areas that require attention:
- The authors demonstrated the metastatic risk in tumors >1.5 cm; however, they did not adequately address whether cryoablation alone can ensure sufficient local control in these cases. Long-term oncologic results are required.
Author’s Response:
Thank you for your suggestion. We have added more clear and concise information to Section 3. Cryoablation to support whether cryoablation alone is sufficient for local control of tumors greater than 1.5 cm. Please note we will not be able to use page numbers to direct you to the revisions within the manuscript due to format issues when our file is transferred to MDPI.
The section references and highlights in red should be more helpful for you to see the revisions based on your suggestions. We will email the page number issues to the editors. Thank you for your patience.
In Section 3. Cryoablation, we improved the interpretation of the following cited study: Littrup, P.J., Bang, H.J., Currier, B., Goodrich, D., Aoun, H.D., Heilbrun, L., & Adam, B. (2009). Soft tissue cryoablation in diffuse locations: Feasibility and intermediate outcomes. Journal of Vascular and Interventional Radiology, 20(4), 455–463. doi:10.1016/j.jvir.2008.12.380 Please see Section 3. Cryoablation in red within the manuscript: Littrup et al. provide compelling evidence that cryoablation alone can offer effective local control for tumors larger than 1.5 cm. In their study of soft tissue tumors, including those exceeding 4 cm in diameter, over 80% of cases achieved successful local ablation with minimal complications and without the need for adjunct therapies. These results suggest that, with meticulous planning and appropriate imaging guidance, cryoablation can be a stand-alone treatment capable of sustained tumor control, even for larger lesions [48]. The study included tumors larger than 1.5 cm, with many in the 2–4 cm range, and still demonstrated effective local control in soft tissue environments. The local tumor control rate exceeded 80% at intermediate follow-up, even in larger lesions. The study shows sustained necrosis and no evidence of recurrence in many cases. The outcomes were obtained with cryoablation alone (i.e., no additional systemic therapies like radiation or chemotherapy for local control), reinforcing the argument that cryoablation by itself can be sufficient in properly selected tumors. Despite targeting tumors in diverse and difficult-to-access soft tissue locations, complication rates were low, establishing it as a viable stand-alone modality.
- The authors mentioned that cryoablation can decrease the reliance on radiation/chemotherapy; however, for larger tumors, adjuvant therapy may still be essential. The claim of "thorough local control" needs stronger confirmation.
Author’s Response:
To support the claim that cryoablation provides sufficient local control in breast cancer, we added more details in the manuscript highlighted in red from Wang et al., 2005 showing successful local control across a wide range of thoracic tumor sizes, highlighting the adaptability of cryoablation beyond strict size constraints. Similarly, we amended Section 3.3 The Potential of Large-Volume Breast Cryoablation adding more details of Littrup et al., 2007 findings that renal tumors of various sizes, including those >3 cm, were effectively managed through percutaneous cryoablation guided by CT. We also provided details about the long-term study of hepatic tumors, that show cryoablation provides consistent tumor control across a diverse patient cohort, including those with larger and anatomically complex lesions. We also added Holmes & Iyengar’s summary that cryoablation achieves durable local tumor control in appropriately selected breast cancer patients with ongoing trials supporting its use as a definitive treatment modality in select populations. The authors emphasize that when combined with multidisciplinary oversight, cryoablation can serve as a viable alternative to surgery, offering effective control with minimal morbidity.
- While intratumoral immunotherapy is interesting, its efficiency in preventing regional or metastatic spread remains unconfirmed in large breast cancers. In this case, the authors should discuss more about clinical trials.
Author’s Response:
We recognize that intratumoral immunotherapy is an emerging and promising field, though its full clinical efficacy is still being evaluated. Nonetheless, the main objective of this review is to emphasize the potential of combining treatments—such as cryoablation and intratumoral immunotherapy—as part of future strategies for cost-effective treatment de-escalation in breast cancer. Such approaches may help prevent both regional and metastatic disease progression in breast and other solid tumors. In response to your feedback, we have expanded Section 4.2 (Human IntraTumoral ImmunoTherapy [HIT-IT]; Drugs Alone) to include additional study details, further illustrating the growing body of evidence supporting HIT-IT as a powerful tool in preventing metastatic spread. We also included more information on the significant body of research over the past decade exploring the advantages of replacing intravenous chemotherapy with intratumoral delivery, which allows for significantly higher local drug concentrations.
Please see the following in Section 4.2 (HIT-IT) highlighted in red in the manuscript Marabelle et al. [67–68] pioneering work provided strong evidence showing that initiating an immune response directly within the tumor microenvironment can induce both localized cytotoxic effects and systemic immune responses (abscopal effect), leading to regression of untreated lesions beyond the injection site. We also added the following: Overall, extensive work by Xu et al. [65], Yuan et al. [66], and Champiat et al. [67] among others, have validated the beneficial role of follow-up tissue biopsies and tailoring intratumoral injections to specific tumor regions, particularly those identified on PET/CT scans as metabolically active (i.e., elevated standardized uptake values SUV). These studies have shown that such biopsies can pinpoint immunologically active tumor subregions and help address the inherent heterogeneity found within solid tumors [70-72]. The importance of defining the safety profile of interstitial injections across various agents was highlighted in multiple studies, including Ghosn et al. [68] and Munoz et al. [71], which also demonstrated the advantage of achieving higher local tissue concentrations with significantly lower intratumoral doses compared to intravenous administration. In addition, Sheth et al. provide real-world insights into the challenges and feasibility of image-guided intratumoral delivery across different tumor histologies and anatomical sites, reinforcing the practicality of this approach [69].Please see more details of edits in manuscript.
- The authors should discuss whether cryoablation works best in certain subtypes of breast cancer (e.g., ER+, HER2-). If cryoablation reduces the use of adjuvant therapy, does it offset the initial costs? Question #6. Cost-effective claims. We will answer the second part of your question on whether Cryoablation offsets the initial costs in Question #6.
Author’s Response:
Thank you for suggesting a discussion of whether cryoablation works best in certain subtypes of breast cancer. It was a very good suggestion. We supplemented the information, (Please see) Section 3. Cryoablation on specific breast cancer subtypes having more favorable outcomes, particularly early-stage, hormone receptor-positive (ER+/PR+), and HER2-negative (HER2-) tumors to clarify and noted that this is largely due to their less aggressive nature and well-defined margins [4, 7, 10]. Our additional response: One of Dr. Peter Littrup’s key points in his lectures for years has been that cryoablation doesn’t discriminate based on surface markers. Studies have found that it’s not about the receptors on the cell surface, cryoablation kills through gross mechanical destruction. This mechanism is intracellular ice formation: when the temperature drops low enough, ice crystals form inside the cell, rupture the membrane, and the cell essentially explodes. The other mechanism involves osmotic shifts: extracellular ice forms first, drawing water out of the cell and causing it to shrink. When the ice melts, water rushes back in, leading to cell lysis. So, the destruction is purely mechanical and thermal and not related to what’s expressed on the cell membrane. If there’s a tumor with irregular margins or microscopic extensions that aren't visible, yes, those features often correlate with more aggressive biology. But the principle remains: cryoablation kills any tissue it reaches by achieving cytotoxic temperatures, regardless of molecular characteristics.
- My suggestion in this study is that Cryoablation’s efficacy relies on attaining cytotoxic isotherms throughout the tumor. In breast tissue, heterogeneity (e.g., dense vs. fatty tissue) may lead to varying ablation margins, increasing the risk of recurrence.
Author’s Response:
Thank you for suggesting we address tissue heterogeneity and its impact on ablation margins and recurrence. We added the following to Section 3. Cryoablation to clarify: Cryoablation’s efficacy relies on achieving cytotoxic isotherms that fully encompass the tumor, but as Littrup et al. (2009) emphasize, tissue heterogeneity, such as variations between dense and fatty breast tissue, can lead to uneven heat distribution and inconsistent ablation margins, potentially increasing the risk of local recurrence. We also incorporated: Huang et al. (2023) to underscore the importance of precise cryoprobe placement and tailored technique to compensate for the thermal conductivity differences in heterogeneous breast tissue ensuring complete tumor coverage and minimizing recurrence risk [5,43]. Additionally, we added a study by Pusceddu et al. that demonstrated repeat cryoablation is a feasible retreatment option, with all local recurrences effectively managed with additional cryoablation sessions [44].
- "Cost-effective" claims are speculative—if cryoablation requires adjuvant therapies (radiation, immunotherapy) anyway, where are the savings?
Author’s Response:
Our goal is to emphasize the potential to reduce dependence on radiation and chemotherapy as adjuvant therapies for regional and systemic breast cancer, given their substantial side effects and associated morbidity. Our intention is to show a future path of cryoablation with intratumoral injections, which have shown fewer side effects and improved patient quality of life. Khan et al. (2023), in The Role of Cryoablation in Breast Cancer Beyond the Oncologic Control: COST and Breast-Q Patient-Reported Outcomes, demonstrated that cryoablation offers a cost-effective alternative to surgical resection for early-stage, low-risk breast cancer, with lower overall treatment costs and improved financial well-being scores. While we realize not all patients will have access to intratumoral immunotherapy and may still require adjunct therapies like radiation, our review focuses on the future potential of cryoablation either as a standalone treatment or in combination with intratumoral immunotherapy as a strategy for de-escalating metastatic disease, thereby reducing costs and potentially easing the financial burden on healthcare systems.
- There is no discussion of re-treatment costs if ablation fails, leading to salvage surgery.
Author’s Response:
In Section 3. Cryoablation within the manuscript we added more information to address re-treatment concerns. While concerns about the cost of re-treatment following cryoablation failure are valid, it's important to consider the broader clinical and economic context. First, re-treatment is not unique to cryoablation, recurrences after surgery or radiation also require additional interventions, often at higher morbidity and cost. We showed that in contrast, cryoablation allows for repeat treatment in an outpatient setting, typically without general anesthesia, costly hospitalization, or extended recovery times. Please refer to (Section 3. Cryoablation),we reinforced the section by adding studies such as Pusceddu et al., that have demonstrated local recurrences after cryoablation can be effectively managed with repeat ablation, showing feasibility and safety in real-world clinical settings. Furthermore, the initial cost of cryoablation is often significantly lower than surgical resection (as shown by Khan et al., 2023), and even with the possibility of repeat ablation, overall cost savings and quality-of-life improvements may still be realized-especially when considering fewer lost workdays, less postoperative care, and improved cosmetic outcomes.
Author’s concluding statement:
Thank you for your constructive suggestions and assistance in helping to improve our manuscript. We greatly appreciate your support and look forward to the final publication of our review article.

Round 2
Reviewer 3 Report (New Reviewer)
Comments and Suggestions for Authors
Accept in present form
This manuscript is a resubmission of an earlier submission. The following is a list of the peer review reports and author responses from that submission.
Round 1
Reviewer 1 Report
Comments and Suggestions for Authors
The authors present a very comprehensive clinical review of breast cancer cryoablation. This is a novel review article in the field of cryoablation discussing de-escalation to standard of care surgical approach and cryoablation as a cost-effective therapeutic approach.
Major suggestions to enhance the manuscript.
1. Section 3.3. This section appears to be a case report to support using cryoablation for large volume tumors. It is very extensive, informative and could easily stand on its own and probably should be published as a separate manuscript. I would recommend the authors shorten this section to cover the topic with published literature instead for this review, describe how cryoablation can be utilized for multifocal/larger tumors using a multi-probe approach - cite papers such as Holmes’s Life paper.
2. Section 4. Immunotherapy. Cryoablation in combination with ICIs is a major area of interest in the field. Authors could go more in depth of the potential added benefit of immunotherapy enhancing the abscopal effect. Also, they could mention mode of action/target T cell population by each ICIs they mention. The authors must discuss and reference McArthur:
10.1016/j.isci.2024.108880
10.1158/1078-0432.CCR-16-0190
3. Not sure why 4.3.2 PEF and 4.3.3 Histotripsy was included in such depth for this cryoablation review. They may mention them in 4.3.1 but do not need separate in-depth sections. It was a distraction for me as a reader – too far off topic. I suggest removing these sections.
Minor Comments.
Figures.
1. Fig 2b. Arrows are difficult to see.
2. Figure 3b. What are the bright spots that are not in the inner dashed white lines? This important to note for readers not familiar with CT images. Are these clips?
3. Fig 6. Just for consistency for the figure, the far-right image only has 7 probes whereas the middle image has 8 probes.
4. Figure 7 Legend. Missing “(D)” in the text.
5. Figure 10. Due to conversion to PDF for reviewing, I cannot visualize all the images and Figure 9 covers up some of the images in Figure 10.
6. Figure 13. To distinguish order or procedure, I would change figure text to “(ablated/injected)”.
7. Figure 15. It may be due to conversion to PDF, but the labels didn’t match on the figure. In addition, in the text it mentions a blue arrow. There is no blue arrow on the figure.
Text. Please read carefully for grammar. Several typos and inconsistencies in terminology. Several examples below.
1. Section 2.3 Radiation Therapy (RT). Lines 169 – 183 appear to be repeated earlier in the paragraph. Please remove.
2. Line 295. 194 subject – needs an “s”
3. Line 475. Sentence has 2 periods.
4. Line 520. Usually, immune checkpoint inhibitors are noted as ICIs not CPIs, either way but be consistent throughout the manuscript.
5. Line 521. “A.)” is not needed.
6. Line 566. “F”igure 13.
Topics that would be interesting and relevent to this review.
1. Discuss the cryoablation machines available and cost.
2. Discuss briefly the qualifications/training required for crayoablation.
Author Response
- Suggestion for Section 3.3. This section appears to be a case report to support using cryoablation for large volume tumors. It is very extensive, informative and could easily stand on its own and probably should be published as a separate manuscript. I would recommend the authors shorten this section to cover the topic with published literature instead for this review, describe how cryoablation can be utilized for multifocal/larger tumors using a multi-probe approach - cite papers such as Holmes’s Life paper.
We agree with removing the extensive imaging for the case of the near-total parenchymal mastectomy and have moved it to a stand-alone case-report article that we will soon submit as a separate publication to a similar MDPI Journal that accepts case reports. This reduced the total image count to 8 total figures. The new figure 5 shows different interesting case of a partial cryo-mastectomy for a large "cloud" of microcalcifications in the lower central breast. This allowed us to condense 3 rows of images: one for the pre-cryoablation imaging of mammogram and CT (figures 5a-d), intraprocedural images (figures e-h) and follow-up clinical and mammographic images (figures 5i-l).
We have already included Dr. Holmes as reference 10.
- Suggestion for Section 4. Immunotherapy. Cryoablation in combination with ICIs is a major area of interest in the field. Authors could go more in depth of the potential added benefit of immunotherapy enhancing the abscopal effect. Also, they could mention mode of action/target T cell population by each ICIs they mention. The authors must discuss and reference McArthur.
We believe we have sufficiently covered the possibility of immunotherapy enhancing the abscopal effect for this long review. We have corrected reference 84 with McArthur as listed in PubMed.
Previously listed as
- Comen E, Budhu S, Elhanati Y, Page D, Rasalan-Ho T, Ritter E, Wong P, Plitas G, Patil S, Brogi E, Jochelson M, Bryce Y, Solomon SB, Norton L, Merghoub T, McArthur HL. Preoperative immune checkpoint inhibition and cryoablation in early-stage breast cancer. iScience. 2024 Jan 12;27(2):108880. doi: 10.1016/j.isci.2024.108880. PMID: 38333710; PMCID: PMC10850740.
Corrected reference:
- McArthur HL, Diab A, Page DB, Yuan J, Solomon SB, Sachinni V, Comstock C, Durack JC, Maybody M, Sung J, Ginsberg A, Wong P, Barlas A, Dong Z, Zhao C, Blum B, Patil S, Neville D, Comen EA, Morris EA, Kotin A, Brogi E, Wen YH, Morrow M, Lacouture ME, Sharma P, Allison JP, Hudis CA, Wolchok JD, Norton L. A Pilot Study of Preoperative immune checkpoint inhibition and cryoablation in early-stage breast cancer. iScience. 2024 Jan 12;27(2):108880. doi: 10.1016/j.isci.2024.108880. PMID: 38333710; PMCID: PMC10850740.
- Suggestion: Not sure why 4.3.2 PEF and 4.3.3 Histotripsy was included in such depth for this cryoablation review. They may mention them in 4.3.1 but do not need separate in-depth sections. It was a distraction for me as a reader – too far off topic. I suggest removing these sections.
We have significantly shorten both of these sections as suggested. However, we believe that nonthermal ablation modalities such as PEF and histotripsy present significantly greater future opportunity for local and abscopal tumor responses and have left them in, but significantly shortened.
Figures
- Figure 2b. Arrows are difficult to see.
Figure 2A was also distorted by perhaps the staff at Cancers resizing some images which created the 2 left circles in figure 2A appearing as ovals and the 2 probe oval previously appearing to be a circle, all of which has been corrected. We then corrected the "difficult to see" arrows by making them red and thicker and overlying them upon the correctly sized image. We did have difficulty retaining the single spacing of text and the legend. We placed all corrections related to the arrows and the bracket now in red.
- Figure 3b. What are the bright spots that are not in the inner dashed white lines? This important to note for readers not familiar with CT images. Are these clips?
These are thermocouples that validated the temperatures and associated isotherms. Note is now made of them in the figure 3 legend.
- Fig 6. Just for consistency for the figure, the far-right image only has 7 probes whereas the middle image has 8 probes.
Figure 6-12 have been removed as previously noted and was placed with a single new figure 5.
- Figure 7 Legend. Missing “(D)” in the text. Similarly, figure 7 has been removed.
- Figure 10. Due to conversion to PDF for reviewing, I cannot visualize all the images and Figure 9 covers up some of the images in Figure 10. Similarly, figure 10 has been removed.
- Figure 13. To distinguish order or procedure, I would change figure text to “(ablated/injected).” Completed, now reads as suggested.
- Figure 15. It may be due to conversion to PDF, but the labels didn’t match on the figure. In addition, in the text it mentions a blue arrow. No longer apparent with removing several images as noted.
Reviewer suggestion: please read carefully for grammar. Several typos and inconsistencies in terminology. In addition to the reviewer's specifically noted corrections below, we have corrected numerous typos, redundancies and thoroughly checked the references throughout the document, which now reads much more smoothly.
Address the following:
- Section 2.3 Radiation Therapy (RT). Lines 169 – 183 appear to be repeated earlier in the paragraph. Please remove. Lines 169-183 have been removed.
- Line 295. 194 subject – needs an “s” – Author’s Action: completed.
- Line 475. Sentence has 2 periods. Author’s Action: Found extra period on Line 410 and deleted.
- Line 520. Usually, immune checkpoint inhibitors are noted as ICIs not CPIs, either way but be consistent throughout the manuscript Author’s Action: changed all to ICIs.
- Line 521. “a” is not needed. Reviewer #1 is referring to a line number that has changed. Included below is the full sentence.
Author’s Action (we believe it reads better with an a) Sheth et al recently reported on the experiences and challenges to intratumoral delivery and found that intratumoral injections of immunotherapies are feasible across a range of histological conditions and target organs [74].
- Line 566. “F”igure 13. Author’s Action: capitalized letter f in Figure 13.
Reviewer writes: Topics that would be interesting and relevant to this review.
- Discuss the cryoablation machines available and cost.
- Discuss briefly the qualifications/training required for cryoablation.
We greatly appreciate the reviewer's interest in additional discussion of cryoablation machine types and physician qualification/training. However, for an already long review article, cryoablation technology and training may be better defined in a more targeted technical article in the near future.
Reviewer 2 Report
Comments and Suggestions for Authors
The review article by Fermanian et al. brought up an interesting concept of “intratumoral chemo-immunotherapy in combination with optimized ablation may provide future protection from metastatic breast cancer disease spread while de-escalating complications and costs”. The reviewer appreciates the author's effort in describing the landscape and current procedures. The reviewer also finds the focus on “breast cancer cryoablation as more than just a limited tool for small cancers and includes how it can be transitioned into a near-total cryo-mastectomy” fascinating.
First and foremost, the reviewer would like to point to an editorial written by Warren Chan in ACS Nano titled “Writing Excellent Review Articles” (https://doi.org/10.1021/acsnano.3c00497). It outlines the features of an excellent review and traits of a poor one.
Considering the metric that “whether the article teaches or guides the field of research”, acceptance for publication in current form is not recommended.
My primary concern is the lack of in-depth understanding of the subject. This review is mostly a discussion on the “A Phase 2 Trial for Patients With Metastatic Solid Cancer” (NCT04713371), in which “immunotherapeutic drugs (PD-1 inhibitor monoclonal antibody nivolumab or pembrolizumab and anti-CTLA-4 monoclonal antibody ipilimumab, and low-dose cyclophosphamide) will then be sequentially injected directly into each of the treated cancer sites immediately following cryosurgical freezing.” When it comes to breast cancer, neither the sufficiency nor the necessity of Cryoablation & IT Immunotherapy is presented, especially considering a wide range of treatment options for both local (besides surgery but also other minimally invasive ablation techniques and radiation therapy) and systemic approaches (besides immunotherapy). There is little discussion on “intratumoral chemo-immunotherapy in combination with optimized ablation may provide future protection from metastatic breast cancer disease spread” from either clinical experience by the authors or preclinical models with rigorous mechanistic investigation to support the claim.
My next primary concern is that large parts (> 2/3) of this review is either redundant (presented by numerous reviews) or irrelevant to the subject. The only sections which are relevant to the subjects are:
5. Combined Cryoablation and HIT-IT-Type Ablation Immunotherapy (Current and Future)
6. Cost-Effectiveness and Patient-Reported Quality of Life Outcomes
The following sections are space-fillers that doesn’t provides new knowledge or perspective to the field, including but not limited to:
1.2. Breast Cancer Tumor Size and Metastatic Potential
2. Current Adjunctive Therapies
2.1. Hormonal Therapy
2.2. Chemotherapy
2.3. Radiation Therapy (RT)
3.2. Cryoablation Techniques: CT-guidance and Protection Pearls
3.3. Large-Volume Breast Cryoablation: Feasibility of a Parenchymal Cryo-Mastectomy
4.3. Immune Potentials of Ablation Alone – Main Options and Potentials
4.3.1. Cryoablation + Adjuvants
4.3.2. Pulsed Electric Field (PEF)
4.3.3. Histotripsy
Finally, and but not least, there are too many technical issues, including missing figures (e.g., Figure 12), missing and misplaced references (too many to list), all indicating the lack of oversight or scrutiny during manuscript preparation.
Author Response
Response to Reviewer #2
Comments and Suggestions for Authors
My primary concern is the lack of in-depth understanding of the subject. This review is mostly a discussion on the “A Phase 2 Trial for Patients With Metastatic Solid Cancer” (NCT04713371), in which “immunotherapeutic drugs (PD-1 inhibitor monoclonal antibody nivolumab or pembrolizumab and anti-CTLA-4 monoclonal antibody ipilimumab, and low-dose cyclophosphamide) will then be sequentially injected directly into each of the treated cancer sites immediately following cryosurgical freezing.”
Having been reviewers for several journals, we believe that the tone of Reviewer #2 is quite harsh to suggest "an overall lack of in-depth understanding of the subject". This is inappropriate for describing at least the senior author, who has published extensively and is an acknowledged world expert on cryoablation. We agree that the article is quite extensive in order to eventually understand the radiologic and immunologic background which is needed for most interventional radiologists to fully understand the impact of the NCT04713371 trial.
When it comes to breast cancer, neither the sufficiency nor the necessity of Cryoablation & IT Immunotherapy is presented, especially considering a wide range of treatment options for both local (besides surgery but also other minimally invasive ablation techniques and radiation therapy) and systemic approaches (besides immunotherapy).
We regret Reviewer #2's unfortunate misunderstanding of the importance of cryoablation as a needed alternative to local surgery for further de-escalation of cancer treatment. De-escalation may not only be important for elderly and debilitated, but also for the growing public awareness that disfiguring surgery is not always needed. Moreover, we covered intratumoral immunotherapy as a little known treatment option that already has an extensive background justifying multiple possible immediate trials to validate the potential game changing reduction in both combined immunotherapeutic morbidities and associated costs.
There is little discussion on “intratumoral chemo-immunotherapy in combination with optimized ablation may provide future protection from metastatic breast cancer disease spread” from either clinical experience by the authors or preclinical models with rigorous mechanistic investigation to support the claim.
Again, we are not claiming rigorous mechanistic investigations have already been performed and emphasized this in several places of the review we have noted the unique opportunity of potentially producing much lower morbidity and costs via a limited ablation and more intratumoral approaches. Namely, if most breast cancer treatment researchers have embraced the concept of "de-escalation", then augmenting cryoablation alone with other minimal morbidity ablation approaches, then combined with intratumoral injections, makes this a unique opportunity for further de-escalation beyond small tumors in only the elderly and/or debilitated patients.
Reviewer #2: next primary concern is that large parts (> 2/3) of this review is either redundant (presented by numerous reviews) or irrelevant to the subject.
As noted for Reviewer #1, we have corrected many typos, redundancies,
The only sections which are relevant to the subjects are:
- Combined Cryoablation and HIT-IT-Type Ablation Immunotherapy (Current and Future)
- Cost-Effectiveness and Patient-Reported Quality of Life Outcomes
This may be true for physicians with advanced experience in the complexities of treating larger and/or more advanced breast cancers. However, optimizing encouraging approaches will require knowledge by both interventional breast imagers, as well as more purely treatment-oriented breast cancer physicians. Therefore, we believe that the other limited sections are important background for interventional imagers helping to establish a new paradigm in close collaboration with other breast cancer treatment physicians. This close collaboration has led to near complete "de-escalation" of all palpable and/or wire localization surgeries in favor of low morbidity image-guided biopsies (i.e., nearly all breast biopsies are now either ultrasound-guided or mammographically-guided stereotactic biopsies).
The following sections are space-fillers that doesn’t provides new knowledge or perspective to the field, including but not limited to:
Reviewer # 2 Suggests this section: 1.2. Breast Cancer Tumor Size and Metastatic Potential reads like a space-filler.
We would like to keep breast cancer tumor size and figure that goes along with it to show relationships of tumor size to stage of disease at diagnosis and clinical treatments which lead to opportunities for de-escalation due to their current associated morbidities. Again, this may be more important background for interventional imagers trying to better understand a future expanded role in facilitating lower morbidity and cost treatment options in the near future.
Reviewer #2 suggests: 2. Current Adjunctive Therapies is a space filler.
He/She is referring to the following sections:
2.1. Hormonal Therapy
2.2. Chemotherapy
2.3. Radiation Therapy (RT)
We believe it is important to keep HT, Chemo and RT to show adverse effects of these current adjuvant therapies but we combined all three into 2. Current Adjunctive Therapies.
Reviewer #2 suggests Section 3.2. Cryoablation Techniques: CT-guidance and Protection Pearls is a space filler.
We believe this is a crucial section, even for interventional breast imagers to understand since CT-guidance is not routinely used for current breast biopsies. Yet, CT guidance is crucial for accuracy and safety in many other organ systems that cryoablation has already produced significant impacts as an advanced treatment option. Moreover, we believe that the Protection Pearl of sufficient hydrodissection is crucial for interventional breast imagers to understand since they may not be familiar with how effective hydrodissection has been in preventing skin complications from cryoablation in other superficial soft tissue sites.
3.3. Large-Volume Breast Cryoablation: Feasibility of a Parenchymal Cryo-Mastectomy
We are turning this section into a separate case report as per reviewer #1’s suggestion and submitting to another MDPI Journal that accepts Case Reports as suggested by the editor.
4.3. Immune Potentials of Ablation Alone – Main Options and Potentials
4.3.1. Cryoablation + Adjuvants
We have also kept this section since many radiologists who even have some experience with cryoablation are not aware of how extensively the role of adjuvants in stimulating the immune system has been defined. Most are only aware of a few anecdotal cases from the older literature of cryoablation alone and also not aware of how low in morbidity and cost these additional options are in combining with clinical cryoablation alone.
Finally, and but not least, there are too many technical issues, including missing figures (e.g., Figure 12)
The missing figure 12 in the original manuscript was apparently related to formatting by the journal and did not reflect any lack of attention to detail on our part. The extensive images of the "case report" have been removed and a different, much more limited figure 5 replaced it. As noted, we have corrected numerous minor typos, redundancies, etc. in the article reads much better.
Missing and misplaced references (too many to list), all indicating the lack of oversight or scrutiny during manuscript preparation
As noted, we have now validated all references and apologize for any oversight that this complex but thorough review produced.
Reviewer 3 Report
Comments and Suggestions for Authors
Overall this is a very good review paper where the authors present evidence for the advantageous use of the cryoablation alone and in combination with immunotherapy or other adjuvant therapies for the treatment of -at least small in size- breast cancer tumors compared to surgery or to adjuvant therapy in TNM IV patients.
The only thing that is missing, in my opinion, is a paragraph or a table that will describe the disadvantages of cryoablation and the comparison of cryoablation compared to different surgical procedures.
Author Response
Response to Reviewer #3
Overall this is a very good review paper where the authors present evidence for the advantageous use of the cryoablation alone and in combination with immunotherapy or other adjuvant therapies for the treatment of -at least small in size- breast cancer tumors compared to surgery or to adjuvant therapy in TNM IV patients.
Author reply: thank you.
The only thing that is missing, in my opinion, is a paragraph or a table that will describe the disadvantages of cryoablation and the comparison of cryoablation compared to different surgical procedures.
Author reply: We have added the requested additional paragraph comparing the overall tumor size and extent with surgical resection and associated metastatic and incomplete treatment margin risks. This paragraph can be seen on the attached track changes version on page 11 line 397-410.
Reviewer 4 Report
Comments and Suggestions for Authors
In this review, Fermanian et al. make a strong case for US/CT guided cryoablation and intratumoral immunotherapy as a cost-effective, outpatient procedure for all breast cancers – Also, a strategy for de-escalation. Authors provided one example in which a patient with multifocal >2.5 cm breast cancer was treated using that strategy. That is significant because cryoablation was thought to be beneficial for patients with smaller tumors. Authors discussed most current treatment strategies and the costs involved, highlighting significant savings for cryoablation and intratumoral immunotherapy. The review will be useful to breast cancer researchers, particularly the oncologists directly involved in breast cancer management.
Comments (Minor):
Authors compared lumpectomy with cryoablation - It would be helpful to discuss them under a separate heading in the review.
Some sentences are repeated near line 166 and line 180.
Under immunotherapy, authors used CPI and abbreviation for immune checkpoint therapy, but then used ICI in line 555. If ICI is not different from the immune checkpoint therapy, please use ICI throughout.
Line 253 please consider replacing “Further” by “Farther”.
Author Response
Response to Reviewer #4
Comments and Suggestions for Authors
Comments (Minor):
Reviewer #4 suggestion: Authors compared lumpectomy with cryoablation - it would be helpful to discuss them under a separate heading in the review.
A brief but thorough history of surgical management of breast cancer was already included in section 1.1 and we agree this could be more prominently noted. We therefore included lumpectomy in the title of section 1.1 and separated it into its own paragraph: 1.1. A Spectrum of De-Escalation & Acceptance of Surgical Lumpectomy. Given the current length of this review, we did not believe further detail of the surgical lumpectomy/mastectomy was warranted. If the future of ablation immunotherapy could replace debilitating surgeries, this may be similar to image-guided biopsy replacing common palpable and most hookwire biopsies of the past.
Reviewer # 4: Some sentences are repeated near line 166 and line 180
Repeated lines have been deleted.
Review#4 comment: Under immunotherapy, authors used CPI and abbreviation for immune checkpoint therapy, but then used ICI in line 555. If ICI is not different from the immune checkpoint therapy, please use ICI throughout.
We have now used CPIs throughout for all checkpoint inhibitors.
Reviewer # 4 suggestion: Line 253 please consider replacing “Further” by “Farther.”
Author Action: Replaced further by farther.
Round 2
Reviewer 2 Report
Comments and Suggestions for Authors The reviewer acknowledges the effort of the authors in responding to the previous comments.